# Galectin-3 captures interferon-gamma in the tumor matrix reducing chemokine gradient production and T-cell tumor infiltration

Monica Gordon-Alonso[1,2], Thibault Hirsch[1], Claude Wildmann[1,2] & Pierre van der Bruggen[1,2]

The presence of T cells in tumors predicts overall survival for cancer patients. However, why most tumors are poorly infiltrated by T cells is barely understood. T-cell recruitment towards the tumor requires a chemokine gradient of the critical IFNγ-induced chemokines CXCL9/10/11. Here, we describe how tumors can abolish IFNγ-induced chemokines, thereby reducing T-cell attraction. This mechanism requires extracellular galectin-3, a lectin secreted by tumors. Galectins bind the glycans of glycoproteins and form lattices by oligomerization. We demonstrate that galectin-3 binds the glycans of the extracellular matrix and those decorating IFNγ. In mice bearing human tumors, galectin-3 reduces IFNγ diffusion through the tumor matrix. Galectin antagonists increase intratumoral IFNγ diffusion, CXCL9 gradient and tumor recruitment of adoptively transferred human CD8[+] T cells specific for a tumor antigen. Transfer of T cells reduces tumor growth only if galectin antagonists are injected. Considering that most human cytokines are glycosylated, galectin secretion could be a general strategy for tumor immune evasion.

---

[1] Ludwig Institute for Cancer Research, de Duve Institute, Université catholique de Louvain, Avenue Hippocrate 74, 1200 Brussels, Belgium. [2] WELBIO, Avenue Hippocrate 74, 1200 Brussels, Belgium. Correspondence and requests for materials should be addressed to M.G.-A. (email: monica.gordon-alonso@bru.licr.org) or to P.v.d.B. (email: pierre.vanderbruggen@bru.licr.org)

Clinical efficacy of immunotherapy is limited by a major hurdle: an immunosuppressive tumor microenvironment[1]. The presence of T cells in the tumor bed is among the best predictors of patient survival[2, 3]. However, T cells poorly infiltrate most tumors, and what halts this infiltration is far from being understood. A few mechanisms by which the tumor could hamper T-cell infiltration have been described: nitration of chemokine CCL2 by reactive nitrogen species[4], increased collagen secretion[5], CCR2+ myeloid-derived suppressor cells,[6] and blocked secretion of chemokine CCL4[7].

T-cell infiltration requires a chemokine gradient that diffuses from the tumor, outlining a T-cell enrolment track. Chemokines CCL2, CCL3, CCL4, CCL5, CXCL9, and CXCL10 have been associated with T-cell infiltration into tumors[8]. Among these, CXCL9 and CXCL10 stand out as their tumor expression correlates with prolonged disease-free survival of patients with colorectal carcinoma and other cancers[9]. These chemokines not only attract activated T cells into the tumor but also prevent tumor angiogenesis[10]. They are produced upon interferon (IFN)γ signaling, CXCL9 being exclusively induced by this cytokine[11].

In mouse models, the CXCL9 produced by tumor cells in response to IFNγ was found responsible for T-cell infiltration[12]. Accordingly, in IFNγ-deficient mice, T cells fail to migrate to tumor sites[13]. PD-1 blockade was reported to enhance the production of IFNγ-inducible chemokines, thereby increasing T-cell infiltration[14]. In agreement, blocking DPP-4, a protease that inactivates these chemokines, improves tumor immunity[15]. On the contrary, epigenetic silencing of CXCL9/10 inhibits T-cell infiltration in human ovarian cancers[16].

Tumor cells surround themselves with an extracellular matrix (ECM) that supports their growth, survival and eventually invasive capacity[17]. By secreting and remodeling the ECM, tumors trigger mechanosignaling pathways that promote cell proliferation and enhance metastasis[18, 19]. Alignment of collagen fibers around the tumor islets cooperates to block T-cell penetration, thereby providing an explanation for the frequent location of T cells at the tumor edges[5]. The ECM is a highly glycosylated structure and altered glycosylation is a frequent characteristic of malignancies. Aberrant glycosylation in tumors usually comprises an increased branching of N-glycans and a higher presence of sialic acid[20]. Proteins and lipids with abnormal glycosylation may form new interactions with lectins, i.e., proteins that bind glycans. These interactions have been reported to promote metastasis and immune evasion[21]. Lectins establish numerous interactions with glycans; each interaction being relatively weak and loosely specific compared with protein–protein interactions[22]. However, the combination of these multiple interactions results in strong binding and has an enormous impact in many biological processes[23].

Galectins are lectins that are produced at high levels in most malignancies[24]. As all galectins are multivalent, either by oligomerization or structurally, galectin binding to glycans is cooperative. Multivalency enables galectins to form webs, known as glycoprotein/galectin lattices. Galectin–glycan binding promiscuity and redundancy make very difficult to attribute specific roles to a particular galectin or glycan moiety. Galectins are ubiquitous and display very different functions depending on their subcellular distribution. Extracellular galectins are often observed both soluble and attached to the glycosylated cell surface. Among galectins, extracellular galectin-3 is known to preferentially bind N-glycans. This interaction can be inhibited using (i) sugars that compete for the carbohydrate recognition domain (CRD) with the natural galectin ligands, such as N-Acetyl-D-Lactosamine (LacNAc) and TetraLacNAc; (ii) sugars that interact at a distant site from the CRD, such as GM-CT-01; and (iii) neutralizing anti-galectin-3 antibodies[25].

Extracellular galectin-3 has pleiotropic roles in tumor progression[24, 26]. It binds VEGF-R2 in the tumor microenvironment, increasing its lifetime on the cell surface and consequently favoring tumor angiogenesis[27]. It also binds glycosylated surface receptors on immune cells, such as NKp30, LAG-3, CD8, T cell receptor (TCR), and integrin LFA-1, restraining their clustering and causing NK and T-cell dysfunction[28–33].

We reason that extracellular galectin-3, secreted by the tumor, may accumulate in the tumor microenvironment by binding to the highly glycosylated ECM. Glycoprotein/galectin-3 lattices could therefore retain glycosylated soluble factors, such as cytokines and in particular IFNγ, limiting their diffusion inside the tumor. Trapping IFNγ in the ECM would reduce the CXCL9/10 gradient, thereby limiting T-cell infiltration in the tumor.

## Results

**Human galectin-3 can bind human glycosylated cytokines**. As galectins bind glycosylated moieties that decorate glycoproteins, we examined whether galectin-3 interacts with glycosylated human cytokines and chemokines. Magnetic beads coated with recombinant human galectin-3 were immersed for 1 h in a solution of human recombinant N-glycosylated IFNγ produced by hamster cells. Beads were subsequently removed with a magnet. The amount of remaining IFNγ in the solution was measured by ELISA. IFNγ was captured by galectin-3-coated beads in a dose-dependent manner and released in the presence of lactose, a sugar known to inhibit glycan/galectin-3 binding (Fig. 1a). N-glycosylated IFNγ, produced by human cells, was also trapped by galectin-3-coated beads (Supplementary Fig. 1a). In contrast, unglycosylated IFNγ, produced by *Escherichia coli*, was not captured by these beads (Fig. 1a). Not only lactose but also other galectin antagonists and anti-galectin-3 antibodies were able to block the capture of N-glycosylated IFNγ by galectin-3-coated beads (Fig. 1b). We conclude that retention of IFNγ by galectin-3 is glycan dependent. Another N-glycosylated cytokine, interleukin (IL)-12, which was available only in its glycosylated form, was also assessed for its ability to bind to galectin-3. Galectin-3-coated beads captured IL-12 in a dose-dependent and glycan-dependent manner (Fig. 1c). Neither human O-glycosylated chemokine CCL5 nor N-glycosylated chemokine CCL1 was captured by galectin-3-coated beads (Supplementary Fig. 1b, c).

We also tested the ability of galectin-3-coated beads to capture cytokines naturally present in a human liquid biopsy. We used the synovial fluid from a patient with rheumatoid arthritis, as these fluids are rich in cytokines and chemokines[34]. Galectin-3-coated beads were immersed in the synovial fluid, with or without lactose so as to estimate the glycan-dependent nature of the galectin binding. The beads partially captured some cytokines, such as IL-2, IL-10, IL-12, and IFNγ, in a glycan-dependent manner, as addition of lactose resulted in cytokine release in the supernatant (Fig. 1d; Supplementary Fig. 1d). We conclude that galectin-3 binds specifically the glycans decorating IFNγ and IL-12 and can capture these cytokines in clinical samples.

**Galectin-3 reduces the diffusion of glycosylated cytokines**. Galectin-3 is known to bind highly glycosylated proteins of the extracellular matrix, such as laminin[35]. We wondered whether galectin-3 could simultaneously associate to the extracellular matrix and to glycosylated cytokines, thereby reducing cytokines diffusion through the matrix.

Following the protocol reported for galectin-1-loading of extracellular matrices[36], we first verified that human galectin-3 was retained by a collagen-I matrix. Next, collagen-I matrices were polymerized in transwell chambers, loaded or not with galectin-3,

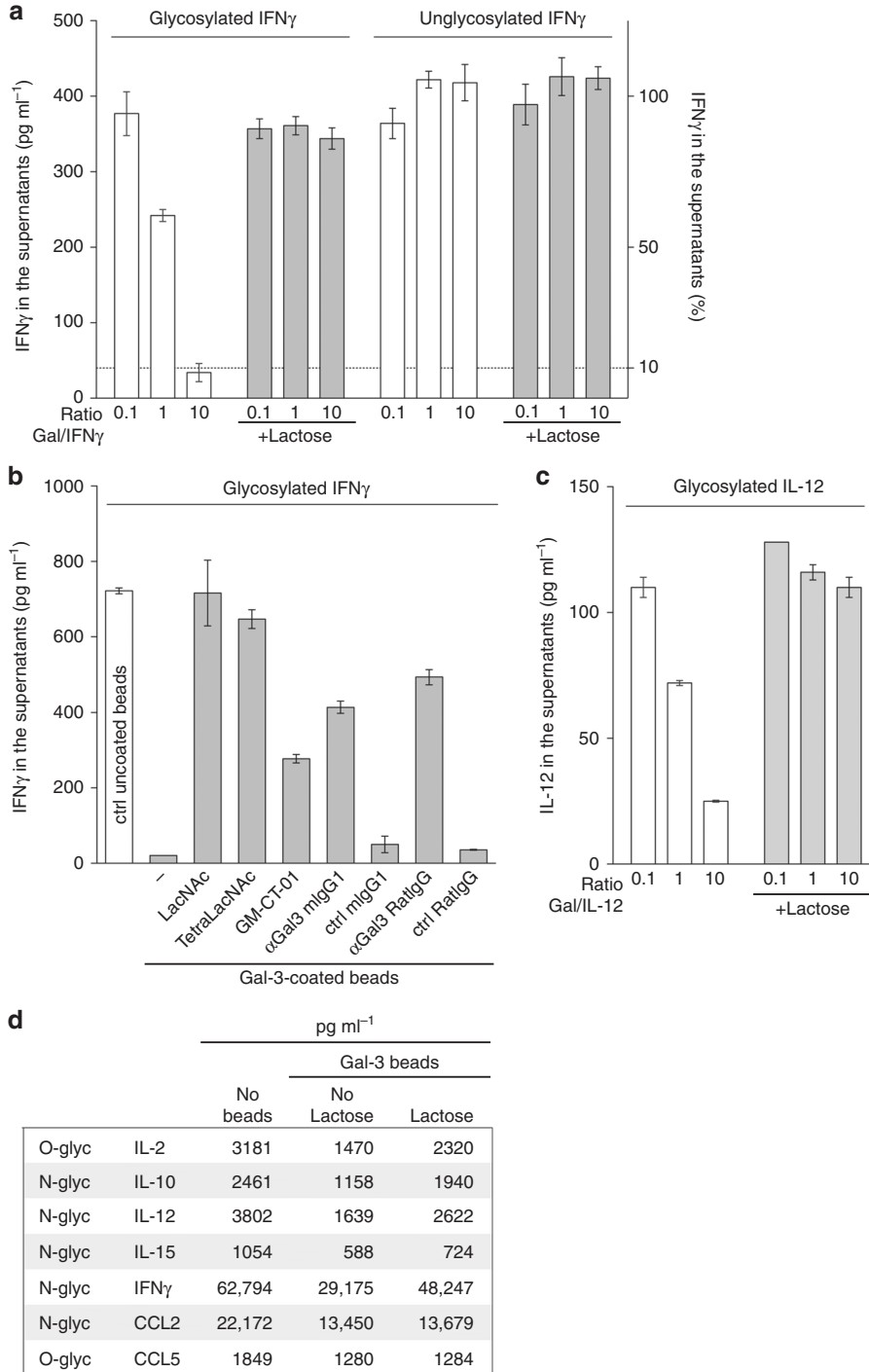

**Fig. 1** Binding of human glycosylated cytokines to galectin-3-coated beads. Ratio refers to molar ratio. **a** hIFNγ measured by ELISA in the supernatant after incubation with galectin-3-coated beads in the presence or absence of 100 mM lactose. The glycosylated IFNγ was produced in CHO cells and the unglycosylated IFNγ was produced in *E. coli*. Mean ± SD of one representative experiment of 8 (glycosylated IFNγ) or 2 (unglycosylated IFNγ), performed in duplicates. The *dotted line* stresses that more than 90% of the glycosylated IFNγ was retained when mixed with 10 times more galectin-3-coated beads. **b** Glycosylated IFNγ measured by ELISA in the supernatant after incubation with galectin-3-coated beads and different galectin antagonists (LacNAc 5 mM, TetraLacNAc 30 μM, GM-CT-01 100 μg ml$^{-1}$), anti-galectin-3 antibody or a control isotype (10 μg ml$^{-1}$). Mean ± SD of one representative experiment of three. **c** Glycosylated hIL-12 measured by ELISA in the supernatant after incubation with galectin-3-coated beads in the presence or absence of 100 mM lactose. Mean ± SD of one representative experiment of three, performed in duplicates. **d** Cytokines and chemokines measured in a human synovial fluid collected from a rheumatoid arthritis patient after incubation with galectin-3-coated beads in the presence or absence of 100 mM lactose. Note: glycosylated cytokines and chemokines were entirely glycosylated, containing no detectable unglycosylated fraction

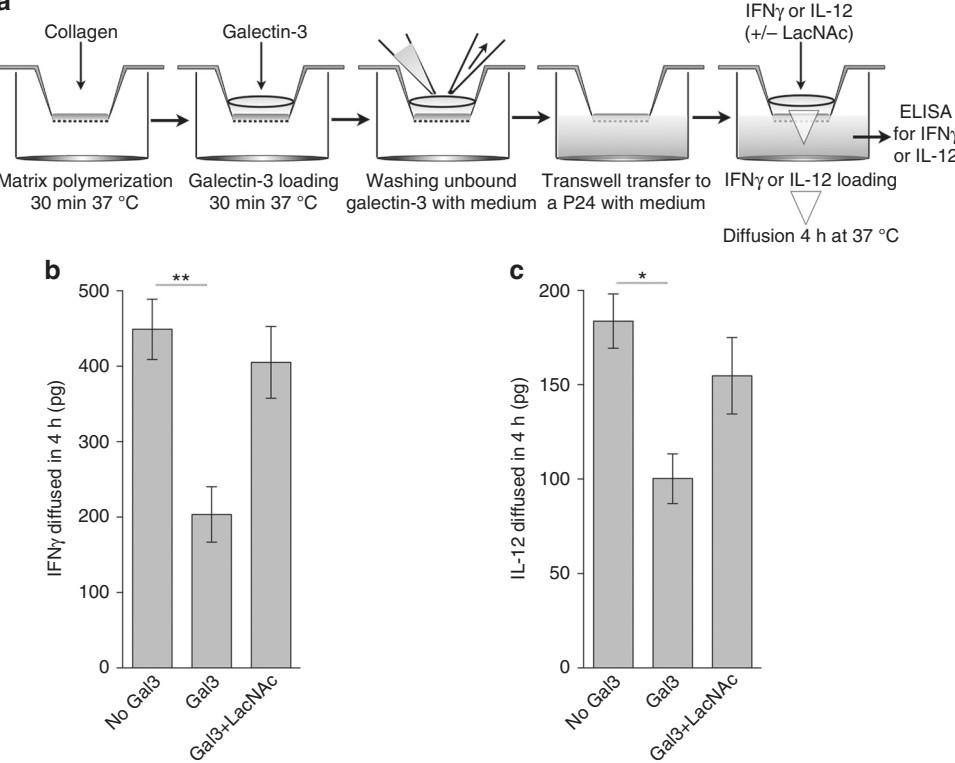

**Fig. 2** Reduced diffusion of glycosylated human IFNγ and IL-12 through galectin-3-loaded collagen matrices. **a** Schematic drawing of the protocol. Galectin-3 (1 μg per transwell), IFNγ (25 ng per transwell), IL-12 (7 ng per transwell), and LacNAc (1.2 mg per transwell). **b** IFNγ diffusion through collagen in 4 h was measured by ELISA. Mean ± SD of three experiments performed in duplicates. $P = 0.006$; Kruskal–Wallis test with Dunn's Multiple Comparison Correction Test. **c** IL-12 diffusion through collagen after 4 h was measured by ELISA. Mean ± SD of three experiments performed in duplicates. $P = 0.032$; Kruskal–Wallis test with Dunn's Multiple Comparison Correction Test

and incubated with glycosylated IFNγ or IL-12 (Fig. 2a). After 4 h, ± 15% of the cytokine had diffused through the collagen-I matrix not loaded with galectin-3 (Fig. 2b, c). When collagen-I matrix was loaded with galectin-3, IFNγ, and IL-12 diffusions were notably reduced (Fig. 2b, c). Cytokine diffusion was restored by addition of galectin antagonist LacNAc (Fig. 2b, c). We conclude that the presence of galectin-3 in collagen-I matrices reduces the diffusion of glycosylated IFNγ and IL-12.

**Galectins reduce IFNγ-induced CXCL9/10 tumor expression.** Signaling through the IFNγ receptor induces the expression of multiple genes in tumor cells. Among them, chemokines CXCL9/10/11 have been proposed as critical because their expression is linked to disease-free survival in cancer patients[9]. We examined whether galectin-3 was able to modulate IFNγ-induced CXCL9 expression in human tumor cells. Human melanoma cell line LB33-MEL secretes galectin-3, which is found both in the supernatant and bound to the cell surface (Fig. 3a). These melanoma cells were treated with glycosylated IFNγ, alone or in combination with galectin antagonist LacNAc. Four hours later, RNA was extracted and CXCL9 and CXCL10 mRNA were measured by RT-PCR (real-time PCR). Addition of LacNAc enhanced IFNγ-induced CXCL9 and CXCL10 expression (Fig. 3b; Supplementary Fig. 2a). In contrast, addition of sucrose, which does not interact with galectins, did not influence CXCL9 expression (Fig. 3b). The LacNAc effect was dependent on IFNγ signaling since a neutralizing IFNγR1 antibody completely blocked CXCL9 induction (Fig. 3b). Noteworthy, PD-L1, a gene induced by IFNγ with immunosuppressive effects, was not significantly enhanced by galectin antagonists (Supplementary Fig. 2b, c).

We had previously shown that galectin-3-covered T cells secreted low levels of IFNγ and that the removal of galectins boosted IFNγ secretion[28–30]. In order to avoid the effect of galectin antagonists on T cells and directly analyze the influence of galectins bound to the extracellular matrix of a solid tumor, we decided to grow human tumor xenografts in immunodeficient mice. NSG mice, which are devoid of T, B, and NK cells, were subcutaneously injected with human LB33-MEL cells. Mice were sacrificed when tumors were bigger than 500 mm³. Tumors were extracted and cut in small pieces of 10–25 mm³. Tumor pieces were immersed for 6 h in media supplemented with glycosylated IFNγ alone or with galectin-3 antagonists. After treatment, the tumor pieces were frozen and CXCL9 mRNA was measured by RT-PCR. IFNγ-induced CXCL9 expression increased in the presence of either LacNAc or an anti-galectin-3 antibody (Fig. 3c). Adding galectin antagonists also boosted IFNγ-induced CXCL10 production in tumor xenografts (Supplementary Fig. 2d). We tentatively conclude that galectin antagonists help the diffusion of IFNγ in the tumor microenvironment by releasing it from the galectin lattices, and thereby increasing CXCL9/10 expression.

We also used two galectin-transfected cell lines derived from the galectin-3 negative human breast tumor cell line SKBR3. The transfected cell lines were expressing either human galectin-3-GFP (SKBR3-Gal3GFP cell line) or a mutated human galectin-3-GFP, which has low affinity for LacNAc (SKBR3-Gal3mutGFP cell line)[37]. Both cell lines showed similar levels of GFP (Supplementary Fig. 3a). Whereas SKBR3-Gal3GFP cells were covered by galectin-3, only a small percentage of SKBR3-Gal3mutGFP cells showed surface galectin-3, most probably because the mutated galectin-3 binds poorly the glycoproteins of the cell surface (Supplementary Fig. 3a). NSG mice were

subcutaneously injected with either of these two cell lines, and tumors were treated ex vivo as described before for LB33-MEL xenografts. In SKBR3-Gal3GFP tumors, galectin-3 antagonists addition enhanced IFNγ-induced CXCL9/10 expression (Fig. 3d; Supplementary Fig. 3b). In contrast, SKBR3-Gal3mutGFP tumors

did not express higher amounts of CXCL9/10 in response to galectin-3 antagonist treatments (Fig. 3e; Supplementary Fig. 3c). All together, these data indicate that the presence of galectin-3 is a limiting factor for IFNγ-induced CXCL9/10 expression by tumor cells.

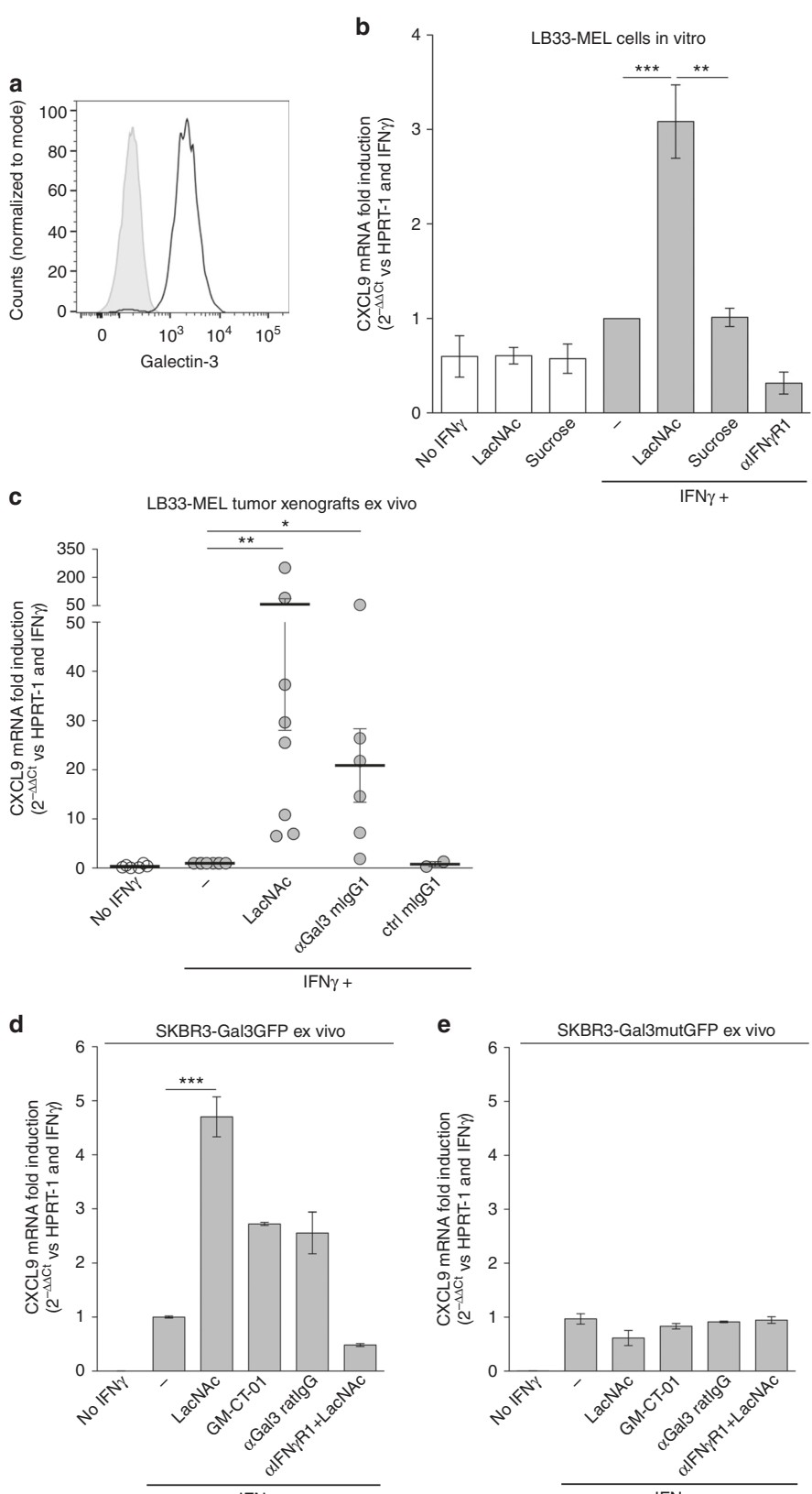

**Galectin-3 inhibition enhances IFNγ function in human biopsies**. Freshly resected human tumor biopsies were obtained from patients with different cancer types (Table 1). Tumors were cut in small pieces and treated ex vivo, as described above for tumor xenografts. Without treatment, CXCL9 basal expression was variable (Table 1, *column untreated, numbers in brackets*). Treatment with a glycosylated IFNγ resulted in CXCL9 expression in most biopsies (Table 1, *column IFNγ*). By adding galectin antagonists to the IFNγ-supplemented medium, we expected to improve the diffusion of IFNγ in a tumor microenvironment supposedly enriched in galectins, thereby increasing CXCL9 expression. In agreement, adding LacNAc enhanced CXCL9 expression by at least two folds in two third of the biopsies (15/23) (Table 1, *column IFNγ + LacNAc*; representative examples in Supplementary Fig. 4a). In a majority of the biopsies responding to LacNAc, other galectin antagonists also boosted CXCL9 expression (Table 1). Noteworthy, in tumors responding to LacNAc, CXCL9 induction correlated with galectin-3 expression (Fig. 4a). Addition of sucrose, which does not interact with galectins, was unable to increase CXCL9 expression (Supplementary Fig. 4a).

A few biopsies may contain secreted IFNγ in their extracellular matrix, explaining the high basal CXCL9 expression found in some of them. In this scenario, treatment with LacNAc alone should release the IFNγ trapped in the matrix, thereby increasing the number of cells expressing CXCL9. Accordingly, LacNAc treatment, without the addition of exogenous IFNγ, increased by at least two folds the basal CXCL9 expression in 57% of the biopsies (12/21; Table 1, *column LacNAc*; representative examples in Supplementary Fig. 4b). Tumors treated with LacNAc only were stratified according to T-cell infiltration, which was estimated by CD3 RNA expression. Interestingly, CXCL9 induction by LacNAc correlated with CD3 presence in tumors with high T-cell infiltration, suggesting that those infiltrating lymphocytes are the source of the galectin-trapped IFNγ, which was released by LacNAc treatment (Fig. 4b). CXCL9 induction resulted from IFNγ signaling, as adding a neutralizing anti-IFNγ receptor antibody blocked it (Supplementary Fig. 4b). We conclude that inhibition of galectins releases the IFNγ trapped in the matrix of human tumor biopsies, facilitating IFNγ diffusion, and consequently increasing CXCL9 expression.

**Galectin antagonists increase intratumoral CXCL9 expression**. To further examine whether galectin-3 retains IFNγ within the tumor microenvironment, we performed in vivo experiments using NSG mice bearing subcutaneous human tumor xenografts. Noteworthy, hIFNγ only signals through human IFNγ receptor and not through its murine ortholog[38]. Thus, in our murine model, only the human melanoma cells can respond to hIFNγ injection. When tumors had reached a volume of 500 mm³, IFNγ was injected in the center of the tumor, alone or in combination with galectin antagonists. Intratumoral injections were performed with 10 μl to minimize structural damages, and trypan blue was added to visualize the injection site afterwards (Fig. 5a;

Supplementary Fig. 5a). Mice were sacrificed one day after intratumoral injection and tumors were frozen. The entire tumor was sliced using a cryostat. Every thirty slices were pooled for RNA extraction and considered as one section (Fig. 5a). CXCL9 mRNA was measured in each section along the whole tumor, as a read-out of IFNγ signal. In the tumors injected with IFNγ and a galectin antagonist, CXCL9 was highly expressed in a large number of tumor sections, producing a gradient that spread up to 5.0 mm from the injection site (Fig. 5b; Supplementary Fig. 6a *for individual tumors*). In contrast, the tumors injected with IFNγ expressed low levels of CXCL9, producing a chemokine gradient up to 0.7 mm from the injection site (Fig. 5b; Supplementary Fig. 6a *for individual tumors*). Injection of galectin antagonists alone did not induce CXCL9 expression. CXCL9 expression at the RNA level was confirmed at the protein level by immunohistochemistry (Fig. 5c; Supplementary Fig. 7). Hence, galectin-3 retains IFNγ in the tumor microenvironment, thereby decreasing the extension of a CXCL9 gradient.

**Collagenase disrupts ECM and enhances CXCL9 expression**. As shown before, recombinant galectin-3 reduces the in vitro diffusion of glycosylated IFNγ and IL-12 through collagen matrices (Fig. 2). To further examine the role of the extracellular matrix in IFNγ trapping by galectin-3, we partially degraded the ECM by injecting in the center of the tumor a low dose of collagenase D together with IFNγ. Tumors were extracted the day after and processed as described before. Adding collagenase D greatly increased the level and area of IFNγ-induced CXCL9 mRNA compared to IFNγ alone (Fig. 6a; Supplementary Fig. 6b *for individual tumors*). Tumors injected with collagenase D alone showed no CXCL9 expression (Supplementary Fig. 8), in agreement with a previous study in a murine syngenic model[39]. CXCL9 expression at the RNA level was confirmed at the protein level by immunohistochemistry (Fig. 6b; Supplementary Fig. 8). Altogether, these data indicate that both the tumor matrix and the extracellular galectin-3 cooperate to limit the spreading of IFNγ signaling.

**Improved tumor T-cell infiltration with galectin antagonists**. We wondered whether an increased CXCL9 gradient upon galectin antagonist treatment was able to attract more T cells into the tumor. NSG mice bearing LB33-MEL subcutaneous tumors were injected intratumorally with IFNγ and galectin antagonists. Two days later, at the peak production of IFNγ-induced chemokines[40], CD8+ T cells loaded with the cell tracker CMFDA were injected intravenously (Fig. 7a). These T cells are autologous to the tumor cells and specific for a mutated antigenic peptide, MUM3, which is expressed by LB33-MEL cells[41]. Mice were sacrificed four days after intravenous T-cell injection. Blood, tumor, and spleen were collected, homogenized and analyzed by flow cytometry to quantify the presence of human CD8+ CMFDA+ T cells (Fig. 7a). The total number of CD8+ T cells recovered from each mouse was variable (between $5 \times 10^4$ and $5 \times 10^5$; Fig. 7b; Supplementary Fig. 9a). CD8+ T cells infiltrating the tumors were more numerous in mice treated with any of the

**Fig. 3** Galectin-3 in human tumors impedes CXCL9 induction by IFNγ. **a** Galectin-3 surface staining of melanoma cell line LB33-MEL. Control isotype staining is depicted in *gray* and galectin staining in *black*. **b** CXCL9 mRNA fold induction in LB33-MEL incubated for 4 h with IFNγ (50 ng ml⁻¹), LacNAc (5 mM), sucrose (5 mM), and/or a neutralizing anti-IFNγR1 antibody (5 μg ml⁻¹). Fold induction CXCL9 values were calculated using HPRT-1 as reference gene and with respect to IFNγ−treated condition ($2^{-\Delta\Delta Ct}$). Mean ± SD of 3–10 experiments. ***P < 0.001 **P < 0.01; Kruskal–Wallis test with Dunn's Multiple Comparison Correction Test. **c** CXCL9 mRNA fold induction in LB33-MEL xenografts treated ex vivo for 6 h with IFNγ (50 ng ml⁻¹), LacNAc (10 mM), and/or different antibodies (10 μg ml⁻¹). Each point represents one tumor. Mean ± SEM of eight independent experiments. **P < 0.01 *P < 0.05; Wilcoxon matched-pairs signed rank test with Dunn's Multiple Comparison Correction Test. **d, e** CXCL9 mRNA fold induction for SKBR3-Gal3GFP (**d**) and SKBR3-Gal3mutGFP (**e**) xenografts treated for 6 h ex vivo with IFNγ (50 ng ml⁻¹), LacNAc (10 mM), GM-CT-01 (100 μg ml⁻¹), and/or antibodies (10 μg ml⁻¹). Mean ± SEM of two independent experiments where pieces were pooled from 12 tumors SKBR3-Gal3GFP and 16 tumors SKBR3-Gal3mutGFP. ***P < 0.001; Wilcoxon matched-pairs signed rank test with Dunn's Multiple Comparison Correction Test

**Table 1 CXCL9 expression in human tumor biopsies treated ex vivo with galectin antagonists.**

| Tumor code | Tumor type | Untreated (*) | LacNAc | IFNγ | IFNγ + galectins antagonists | | | |
| --- | --- | --- | --- | --- | --- | --- | --- | --- |
| | | | | | LacNAc | GM-CT-01 | αGal3 mIgG1 | αGal3 ratIgG |
| 13-121 | Neuroendocrina | 1 (3) | 51 | 317 | 281 | N/A | N/A | N/A |
| 14-031 | Adenocarcinoma | 1 (12) | 1 | 5 | 2 | 2 | N/A | N/A |
| 13-212 | Adenocarcinoma | 1 (33) | 15 | 7 | 82 | N/A | N/A | N/A |
| 14-286 | Renal carcinoma | 1 (67) | N/A | 36 | 91 | | 242 | N/A |
| 14-581 | Pancreatic carcinoma | 1 (77) | 29 | 112 | 392 | 251 | 31 | 8 |
| 13-248 | Adenocarcinoma | 1 (78) | 2 | 1 | 3 | N/A | N/A | N/A |
| 13-480 | Adenocarcinoma | 1 (89) | 0 | 2 | 73 | 12 | N/A | N/A |
| 15-426 | Uterine carcinoma | 1 (139) | 2 | 2 | 5 | 10 | 1 | 1 |
| 12-477 | Adenocarcinoma | 1 (168) | 1 | 1 | 2 | N/A | N/A | N/A |
| 13-298 | Adenoma | 1 (193) | 2 | 2 | 5 | N/A | N/A | N/A |
| 13-568 | Hepatoadenoma | 1 (268) | 1 | 5 | 11 | 9 | N/A | N/A |
| LB-3422 | Melanoma | 1 (379) | 0 | 1 | 16 | 2 | N/A | N/A |
| 14-063 | Adenocarcinoma | 1 (399) | 5 | 7 | 14 | 14 | N/A | N/A |
| 14-583 | Adenocarcinoma | 1 (406) | N/A | 2 | 7 | 2 | 4 | 8 |
| 15-427 | Adenocarcinoma | 1 (489) | 1 | 3 | 3 | 2 | 3 | 2 |
| 15-228 | Sarcoma | 1 (972) | 2 | 98 | 193 | 195 | 116 | 138 |
| 14-187 | Rectal adenocarcinoma | 1 (1065) | 1 | 1 | 5 | 3 | N/A | N/A |
| 13-420 | Lymphoma B diffuse | 1 (1278) | 2 | 2 | 2 | N/A | N/A | N/A |
| 14-526 | Renal carcinoma | 1 (1287) | 2 | 3 | 3 | 3 | 1 | 2 |
| 13-454 | Adenocarcinoma | 1 (1287) | 1 | 2 | 1 | N/A | N/A | N/A |
| 12-529 | Hepatoblastoma | 1 (1628) | 4 | 3 | 8 | N/A | N/A | N/A |
| 14-239 | Renal carcinoma | 1 (3321) | 1 | 2 | 2 | 1 | 1 | N/A |
| 12-549 | Leiomyoma | 1 (8056) | 14 | 5 | 33 | 40 | N/A | N/A |

Fold induction CXCL9 values using HPRT-1 as a reference gene and with respect to the untreated condition ($2^{-\Delta\Delta Ct}$). For the untreated condition, CXCL9 mRNA levels normalized to HPRT-1 ($10^3 \times 2^{-\Delta Ct}$) are indicated in brackets. Human biopsies were treated for 6 h with IFNγ (50 ng ml$^{-1}$), LacNAc (10 mM), GM-CT-01 (100 µg ml$^{-1}$), and/or anti-galectin-3 antibodies (10 µg ml$^{-1}$)
*N/A* no available data

galectin antagonists, whereas the number of CD8$^+$ T cells in the spleens did not vary with the treatment (Fig. 7b; Supplementary Fig. 9a). These data were confirmed by RT-PCR for hCD3ε and hCD8α in tumor and spleen homogenates (Supplementary Fig. 9b). Thus, galectin antagonist treatment changed the distribution of T cells within the mice, enhancing T-cell local retention in the tumor preferentially to the spleen (Fig. 7c). To further examine which chemokines were responsible for the enhanced T-cell infiltration, the experiment was repeated using a specific chemical inhibitor for CXCR3, AMG478[42], in order to block the migration driven by its ligands CXCL9/10/11. The increased tumor T-cell infiltration driven by anti-galectin-3 antibody was completely abrogated by AMG478 administration without affecting the number of T cells found in the spleens (Fig. 7d; Supplementary Fig. 9a). We therefore conclude that a treatment with galectin antagonists improves T-cell infiltration in the tumor, most probably because of a higher production of IFNγ-induced chemokines.

**Galectin-3 inhibition favors the activation of TILs**. To evaluate the activation status of the T cells isolated from tumors and spleens, some markers were analyzed by flow cytometry. Such data are often expressed as percentage of cells positive for a marker, without taking into account the expression level. To convey both parameters, our data were expressed as the percentage of cells positive for the marker multiplied by the median fluorescence intensity of the positive cells.

First, we analyzed the surface expression of CXCR3, the receptor for CXCL9/10/11. CXCR3 was high on spleen T cells and low on tumor T cells independently of the injected treatment (Supplementary Fig. 10a). We have observed in vitro that CXCR3 is downregulated from the surface of anti-MUM3 CD8$^+$ T cells upon antigen recognition or chemokine presence (Supplementary Fig. 10b), in agreement with previously reported data[43]. Therefore, tumor infiltrating T cells seem to have encountered the antigen and/or the chemokine.

Second, we measured the T-cell activation status by staining intracellular IFNγ and granzyme B, and surface expression of CD137 (4-1BB) and CD279 (PD-1). CD137 and CD279 are considered as markers of naturally occurring tumor-reactive T cells present in human tumors[44, 45]. Spleen T cells were mostly resting, showing low intracellular IFNγ, intermediate intracellular granzyme B, no CD137 and low PD-1 (Fig. 7e; Supplementary Fig. 11). All these markers were highly expressed by tumor infiltrating T cells, suggesting that they have encountered their antigen in the tumor (Fig. 7e; Supplementary Fig. 11). Remarkably, whereas spleen T cells show similar activation status independently of the treatment, tumor T cells were more activated when mice were treated with IFNγ and galectin antagonists (Fig. 7e; Supplementary Fig. 11). Hence, adding galectin antagonists results in a higher activation of the tumor infiltrating T cells.

**Galectin-3 inhibition delays tumor growth**. We wondered whether the increased T-cell infiltration and activation phenotype observed in tumors injected with IFNγ and galectin antagonists would have an impact on tumor growth. NSG mice bearing subcutaneous LB33-MEL tumors were treated 15 days after tumor injection, when tumors were 100–150 mm$^3$ (Fig. 8a). The treatments were IFNγ together with either a control isotype antibody or a mix of LacNAc and anti-galectin-3 antibody. On the same day, ten millions of autologous anti-MUM3 CD8$^+$ T cells were injected together with hIL-2 through the tail vein (Fig. 8a). Noteworthy, tumors were compulsory that large at the treatment day so as to be able to inject them intratumorally, which is unusual compared to most therapeutic adoptive cell transfer experiments described in the literature. Injected hIL-2 disappeared within 24 h from the blood and the urine of NSG mice and noticeably, the human anti-MUM3 T cells do not survive in vitro for long in the absence of hIL-2. These remarks could explain why tumor growth was identical between untreated mice and mice treated intratumorally with IFNγ and transferred

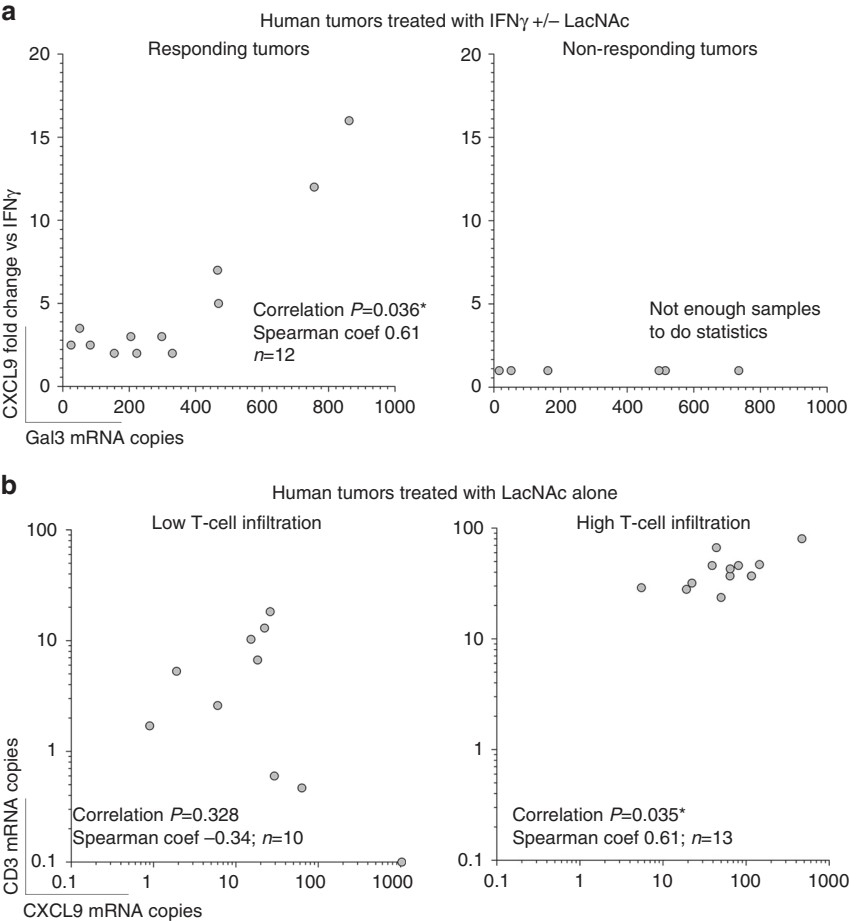

**Fig. 4** Correlation between CXCL9 induction and T-cell infiltration or galectin-3 expression in human tumor biopsies treated ex vivo with galectin antagonists. Samples corresponding to those of Table 1. Correlation probabilities and non-parametric correlation coefficients are shown in each graph. **a** Correlation between CXCL9 induction (fold change induction in samples treated with IFNγ and LacNAc versus their corresponding samples treated with IFNγ alone) and galectin-3 expression in responding or non-responding tumors (having defined responding tumors such as those were CXCL9 fold change was at least two). **b** Correlation between CXCL9 and CD3 expressions in tumors treated with LacNAc alone and stratified depending on their T-cell infiltration. Tumors were considered as highly infiltrated if their CD3 expression was bigger than the average CD3 expression of all the tumor samples

with T cells (Fig. 8b; Supplementary Fig. 12 for *individual tumors*). However, when IFNγ treatment was combined with galectin antagonists, tumor growth was delayed (Fig. 8b; Supplementary Fig. 12, *for individual tumors*). To assess whether galectin-3 inhibition was sufficient to delay the tumor growth, another experiment was performed treating the mice either with IFNγ and anti-galectin-3 antibody or with the antibody alone (Fig. 8c; Supplementary Fig. 12, *for individual tumors*). Anti-galectin-3 antibody was able to delay tumor growth only when combined with IFNγ (Fig. 8c; Supplementary Fig. 12, *for individual tumors*). Taking together these data, we conclude that, in this mouse model, inhibition of galectin-3 reduces the tumor growth by boosting IFNγ intratumoral activity.

## Discussion

We provide here the first proof of principle that a tumor-secreted lectin can modulate the availability of glycosylated soluble factors in the tumor ECM. This study shows that tumor-secreted galectin-3 hijacks IFNγ in the tumor microenvironment, reducing the induction of a chemokine gradient and thereby T-cell recruitment into the tumor bed. Importantly, IFNγ can be released upon treatment with galectin antagonists, resulting in more T cells locally retained at the tumor site and in a delayed tumor growth. Our preliminary data show interaction of galectin-

3 with IL-12 as well. However, further studies are needed to test the effect of various galectins on other glycosylated cytokines, and more importantly, to assess their biological significance.

IFNγ-inducible chemokines CXCL9 and CXCL10 coordinate the recruitment and the fine positioning of anti-tumor immune cells and their production has been correlated with T-cell infiltration in human and murine tumors[9, 12, 39]. This could partly explain why IFNγ was pointed out as essential for T-cell migration toward the tumor in murine models and in human immunotherapy[13, 14]. This work stresses once more the importance of cytokine glycosylation, which regulates not only their potency and lifetime[46], but also, as shown here, their retention in a galectin-containing ECM.

The tumor ECM is a key factor for tumor progression[47]. Dense collagen-I deposition in tumors is an obstacle for tumor infiltration by immune cells[5, 39]. Interestingly, a positive feedback loop has been suggested between galectin-3 secretion, macrophage activation, and collagen-I deposition[48]. Galectin-3 secretion may therefore favor collagen-I deposition by the tumor, contributing to create a dense ECM, and thereby blocking T-cell entry into tumor islets[5].

Our data indicate that the tumor ECM has an essential role in the retention of IFNγ by galectin-3, as collagenase D treatment resulted in increased IFNγ availability and CXCL9 gradient formation. We believe that collagen degradation disrupts the galectin lattices anchored in the ECM, and thereby fade the cooperative binding of

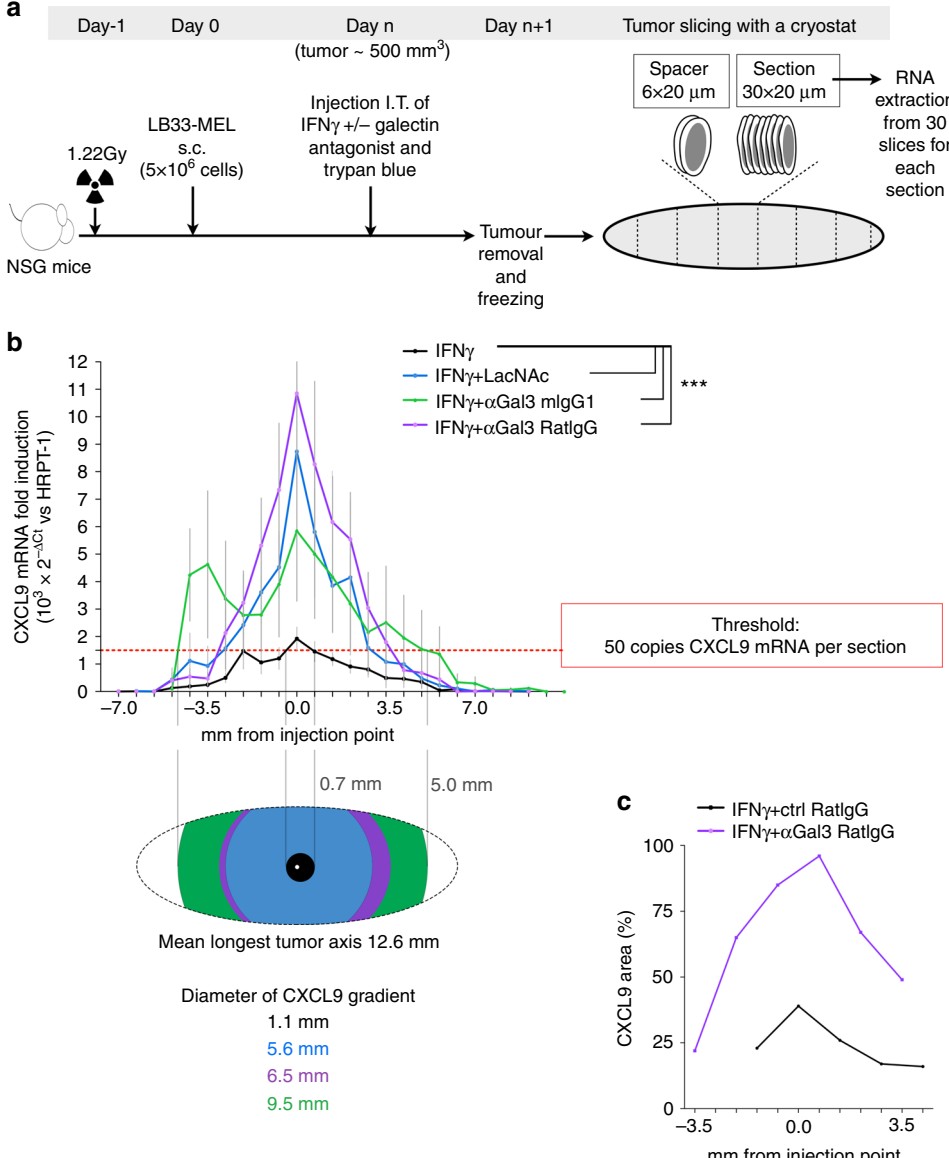

**Fig. 5** Spreading of IFNγ signaling in tumor xenografts treated in vivo with galectin antagonists. **a** Scheme showing the protocol for analyzing IFNγ signal diffusion along the tumor. s.c. stands for subcutaneous injection and I.T. for intratumoral injection. **b** CXCL9 fold induction along the tumor sections. Mice treated with IFNγ alone (50 ng per tumor), or together with galectin antagonists LacNAc (0.1 μmol per tumor) or antibodies (100 ng per tumor). Mean ± SEM of eight independent tumors for each treatment. For CXCL9 induction in individual tumors see Supplementary Fig. 6a. ***P < 0.0001; Wilcoxon matched-pairs signed rank test with Dunn's Multiple Comparison Correction Test. The *red dotted line* marks the threshold of 50 CXCL9 mRNA copies per section of the tumor. Each section represents 0.66 mm thick, contains 30 tumor slices, and about 1–4 millions cells. The drawing below shows the diffusion of CXCL9 gradient along the tumor taking into account the threshold. **c** Quantification of CXCL9 staining area in immunohistochemistry images of several sections of tumors treated with IFNγ and either a control isotype antibody or an anti-galectin3 ratIgG antibody. Representative images are shown in Supplementary Fig. 7

galectins to IFNγ. The absence of an ECM in our in vitro experiments could explain the mild CXCL9 induction observed after treatment with galectin antagonists. Galectin antagonists increased strongly CXCL9 expression in tumor xenografts and human biopsies, supporting the notion that galectin-3 lattices in the tumor ECM are the important players of IFNγ retention. We cannot exclude a synergy between different galectins in cytokine retention in the tumor microenvironment, as LacNAc inhibits several galectins. However, antagonizing only galectin-3 with a specific antibody is sufficient to increase CXCL9 expression.

Other authors have analyzed the general role of galectin-3 using syngeneic and knockouts mouse models and/or systemic administration of galectin antagonists[31, 32, 49–51]. They have shown that

galectin-3 is involved in tumor adhesion, tumor metastasis, tumor angiogenesis, and tumor immune evasion. However, considering that intracellular and extracellular galectin-3 are involved in many processes[22], we have designed our experiments to examine the specific role of extracellular galectin-3 lattices in the tumor ECM. We have therefore used an immunocompromised mouse model bearing a human tumor and injected an anti-galectin-3 antibody locally inside the tumor. A role of intracellular galectin-3 can be disregarded because anti-galectin antibodies are cell-impermeable and the high polarity of LacNAc severely limits its ability to permeate cell membranes[52]. The described roles for extracellular galectin-3 were so far related to surface receptor mobility. Extracellular galectin-3 has been reported to confine IFNγ receptor in

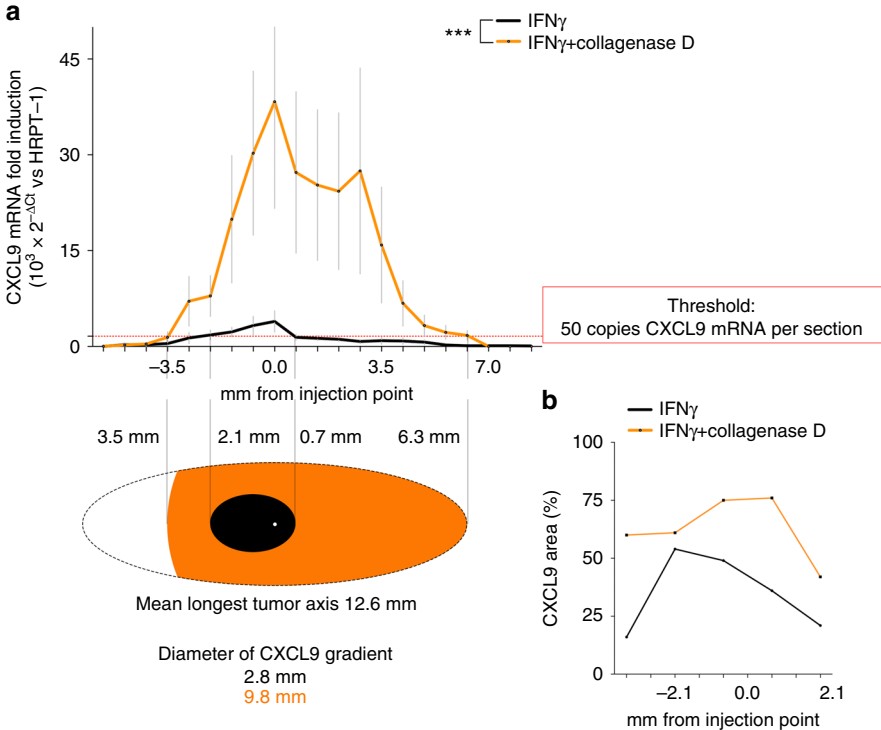

**Fig. 6** Spreading of IFNγ signaling in tumor xenografts treated in vivo with collagenase D. **a** CXCL9 fold induction along the tumor sections in mice treated with IFNγ alone (50 ng per tumor) or together with collagenase D (2.5 µg per tumor). Mean ± SEM of six independent tumors for each treatment. For CXCL9 induction in individual tumors see Supplementary Fig. 6b. ***$P = 0.0006$; Paired $t$-test, $t = 4.035$ and degree of freedom = 20. The *red dotted line* marks the threshold of 50 CXCL9 mRNA copies per section of the tumor. Each section represents 0.66 mm thick, contains 30 tumor slices, and about 1–4 millions cells. The drawing below shows the diffusion of CXCL9 gradient along the tumor taking into account the threshold. **b** Quantification of CXCL9 staining area in immunohistochemistry images of several sections of tumors treated either with IFNγ alone or combined with collagenase D. Representative images are shown in Supplementary Fig. 8

membrane nanodomains, thereby preventing IFNγ signaling[53]. In the in vitro experiments however, retention of IFNγ in galectin-3-loaded collagen-I matrices can only be explained by a direct galectin-cytokine interaction inside the matrix. In vivo the extracellular galectin-3 may retain IFNγ in the ECM and simultaneously prevent IFNγ signaling by confining the IFNγ receptor on the surface of human cells.

We have previously shown another role for extracellular galectin-3 that is related to surface receptor mobility: its ability to impede the lytic activity and the secretion of cytokines of galectin-covered human tumor-infiltrating lymphocytes[28–30]. We wanted therefore to exclude a possible side effect of galectin antagonists on IFNγ secretion by T cells. This was done by injecting human tumor cells in immunodeficient mice devoid of T, B, and NK cells. Another reason to use human xenografts in immunodeficient mice was that our preliminary data indicated that interactions between murine galectin-3 and murine IFNγ were weak compared with human galectin-3 and IFNγ.

As CXCL9 and CXCL10 tumor expression correlates with increased disease-free survival in colorectal cancer-resected patients[9], blocking galectin-3 seems an appealing therapeutic strategy for cancer patients. We believe that the delay in tumor growth observed in the mice treated with galectin antagonists is clinically relevant. Most tumor growth experiments are performed in syngenic murine models with an advantageous set up where adoptive cell transfer is performed soon after tumor injection or even in a prophylactic fashion. In contrast, our humanized murine model and the procedure we have followed is close to the real situation of patients in the clinic, where patients arrive to be treated once the tumors have developed for a certain period. Treating big tumors with autologous human T cells that do not survive for long inside the mice makes very difficult to observe an effect, as illustrated by the fact that adoptive cell transfer and intratumoral IFNγ injection was not able to reduce the tumor growth at all. However, the same strategy plus galectin antagonist did reduce the tumor growth.

Several galectin-3 pro-tumor effects may coexist in human tumors (Fig. 9). For instance, galectin-3 retention of glycosylated cytokines in the tumor ECM on the one hand, and T-cell dysfunction due to galectin-3 on the other hand, as suggested by a more activated T-cell phenotype observed after intratumoral injection of galectin antagonists. Therefore, blocking galectin-3 may have synergic additional anti-tumor side effects, such as e.g., decreasing tumor cell migration[54], collagen-I deposition,[48] and VEGF-driven angiogenesis[27] (Fig. 9). Interestingly, by favoring T-cell infiltration in the tumor bed, galectin antagonists would most probably act in synergy with checkpoint blockade treatments, whose efficacy correlates with tumor T-cell infiltration[55]. This study brings the new concept of tumor-secreted lectins reducing the availability of glycosylated soluble factors in the tumor microenvironment. Further studies are clearly required to test the translational potential of these findings. Our observations suggest that local galectin inhibition during cytokine secretion is needed to boost the cytokine function. We have already observed that some highly infiltrated human tumors contained galectin-retained IFNγ that was effectively released by short ex vivo treatment with galectin antagonists. The effect of longer treatments and systemic administration of galectin antagonists for liberating galectin-retained cytokines has yet to be proven.

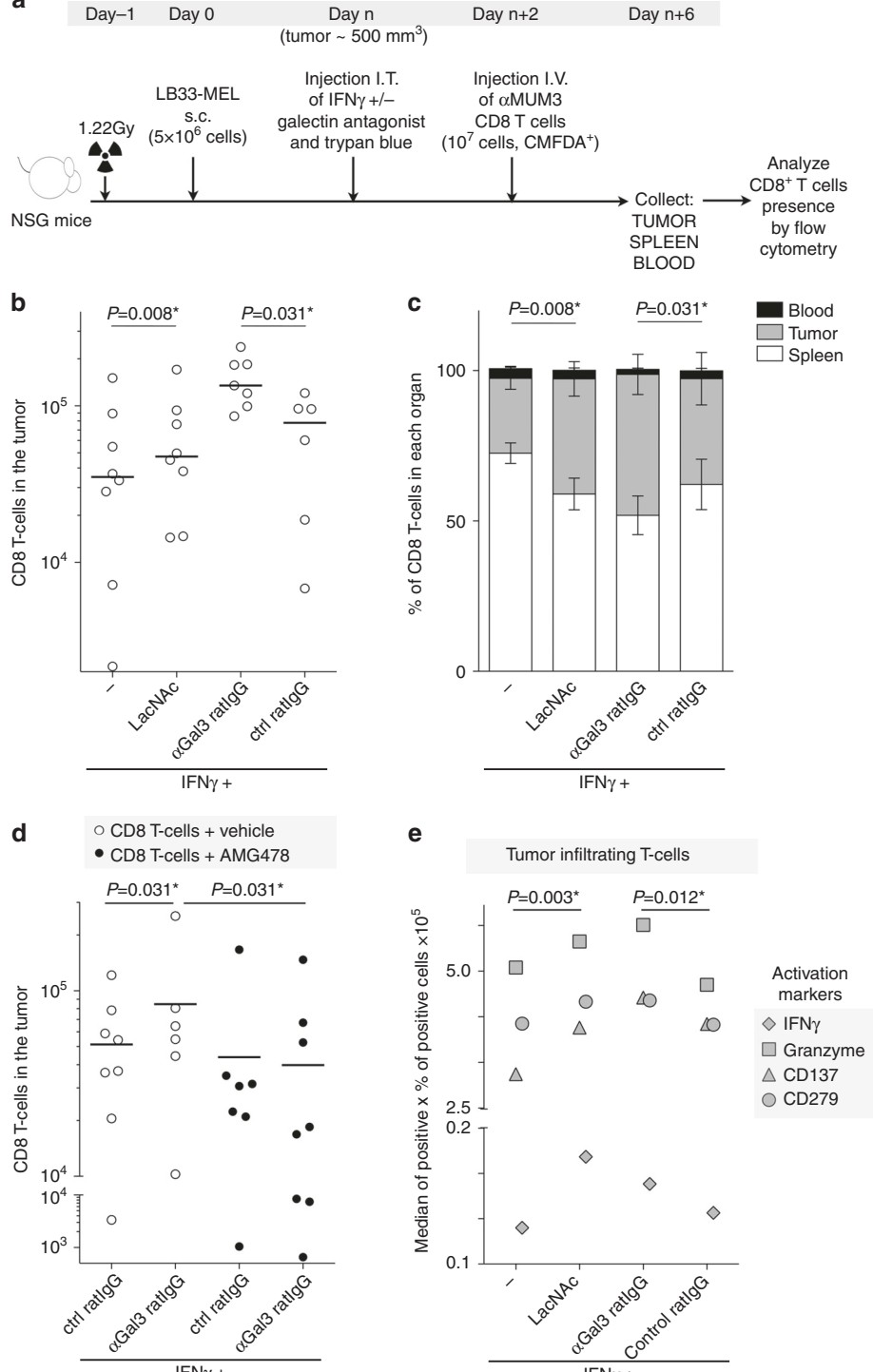

**Fig. 7** Anti-galectin treatments boost CD8[+] T-cell infiltration in the tumor. **a** Scheme showing the protocol for analysis of T-cell infiltration in the tumor. s.c. stands for subcutaneous injection, I.T. for intratumoral injection, and I.V. for intravenous injection. The doses used for I.T. were IFNγ 50 ng per tumor, LacNAc 0.1 μmol per tumor, and antibodies 100 ng per tumor. **b** Absolute numbers of CD8[+] T cells infiltrating the tumors of mice treated as explained above. Each point represents one mouse (average of quadruplicates). *Lines* represent the mean value for each treatment. Wilcoxon matched-pairs signed rank test. *P*-values are shown in the graph. Absolute numbers of CD8[+] T cells in the spleens are shown in Supplementary Fig. 9a; *n* = 6–8 mice per group. **c** Percentage of CD8[+] T cells found in the different compartments. *Bars* represent the mean ± SEM of 6–8 mice per group. Wilcoxon matched-pairs signed rank test. *P*-values regarding tumor and spleen proportions are shown in the graph. **d** Absolute numbers of CD8[+] T cells infiltrating the tumors of mice treated or not with the CXCR3 inhibitor AMG478. Each point represents one mouse (average of quadruplicates). *Lines* represent the mean value for each treatment. Wilcoxon matched-pairs signed rank test. *P*-values are shown in the graph. Absolute numbers of CD8[+] T cells in the spleens are shown in Supplementary Fig. 9a; *n* = 6–8 mice per group. **e** Global activation status of tumor infiltrating CD8[+] T cells. Each symbol represents the mean value for the different activation markers (*n* = 6–8 mice per group). Wilcoxon matched-pairs signed rank test. *P*-values are shown in the graph. Representative histograms are shown in Supplementary Fig. 11

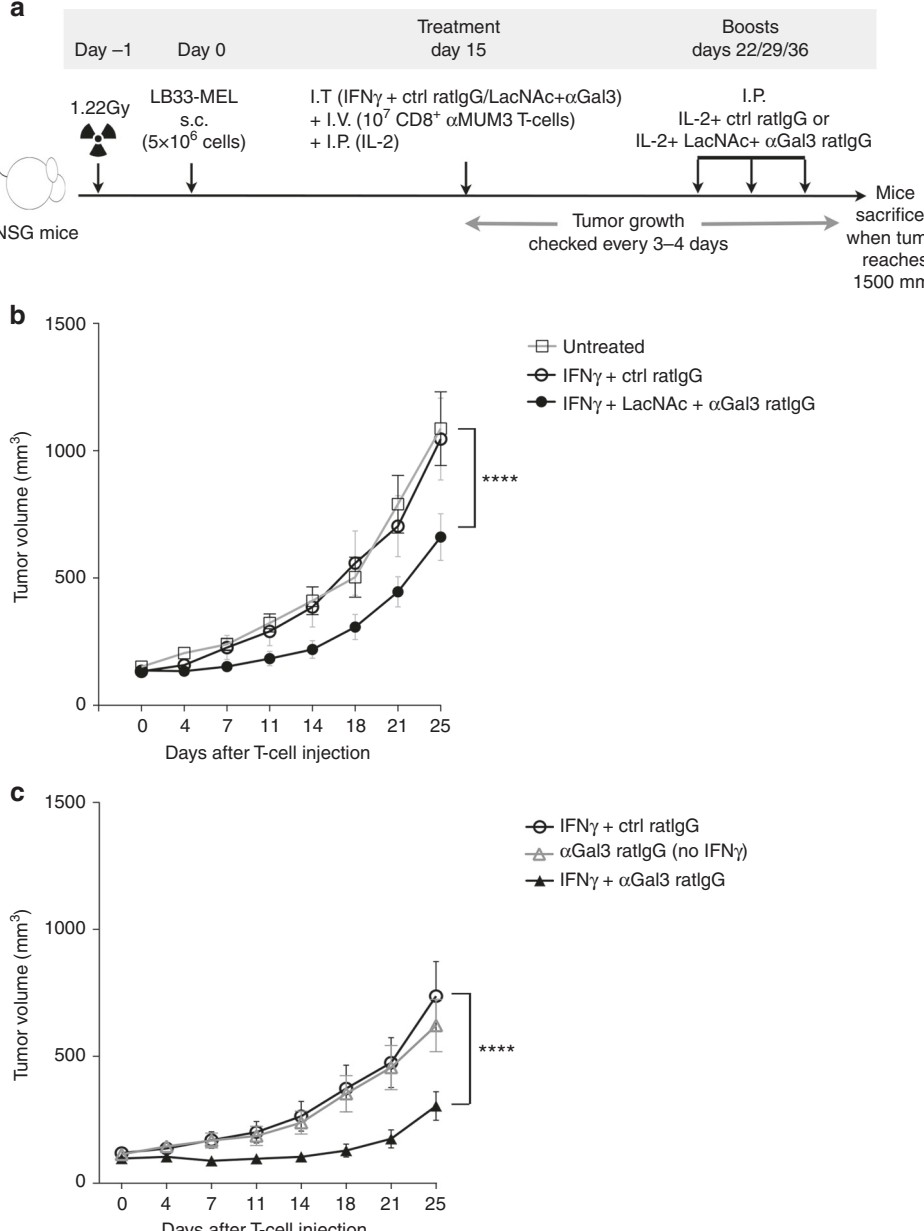

**Fig. 8** Galectin antagonists delay tumor growth. **a** Scheme showing the protocol for the analysis of tumor growth. s.c. stands for subcutaneous injection, I.T. for intratumoral injection, I.V. for intravenous injection, and I.P. for intraperitoneal injection. The doses used for the I.T. treatment were IFNγ 50 ng per tumor, LacNAc 0.1 μmol per tumor, and antibodies 100 ng per tumor. The doses used for the I.P. boosts were IL-2 200 ng per mouse, LacNAc 40 μmol per mouse, and antibodies 50 μg per mouse. **b** Tumor growth in mice untreated or treated with either IFNγ and a control isotype antibody or IFNγ + LacNAc + αGal-3 antibody ($n = 7$, $n = 10$, and $n = 11$, respectively; mean ± SEM). Two-way ANOVA ****$P < 0.0001$, where treatment has $F = 19.83$, and degree of freedom = 1, and time has $F = 24.91$, and degree of freedom = 7. Individual tumor growths are shown in Supplementary Fig. 12. **c** Tumor growth in mice treated with IFNγ and a control isotype antibody, IFNγ and αGal-3 antibody or only with the αGal-3 antibody ($n = 17$, $n = 18$, and $n = 10$ respectively; mean ± SEM). Two-way ANOVA ****$P < 0.0001$, where treatment has $F = 40.26$, and degree of freedom = 1, and time has $F = 13.52$, and degree of freedom = 7. Individual tumor growths are shown in Supplementary Fig. 12

## Methods

**Cells and reagents**. All cell lines were periodically checked for mycoplasma contamination using VenorGeM PCR-based detection kit (Sigma-Aldrich, MP0025). Human cell lines LB33-MEL, LB33-EBV, and anti-MUM3 CD8 T cells were derived in 1988 from melanoma patient LB33[41]. Melanoma LB33-MEL and LB33-EBV B cell lines were cultured in Iscove Medium (Gibco; #12440053) complemented with 10% FBS (Sigma; F7524), 0.55 mM L-Arginine, 0.24 mM L-Asparagine, 1.5 mM Glutamine (AAG), 0.75 mM β-Mercaptoethanol, 100 U ml⁻¹ penicillin, and 100 μg ml⁻¹ streptomycin. Human breast tumor SKBR3-derived cell lines were a kind gift from Emma Salomonsson and Hakon Leffler and were cultured in RPMI1640 (Gibco; #52400-025) complemented with 10% FBS (Sigma; F7524), 0.55 mM L-Arginine, 0.24 mM L-Asparagine, 1.5 mM Glutamine (AAG),

1 mM sodium pyruvate, 100 U ml⁻¹ penicillin, and 100 μg ml⁻¹ streptomycin[37]. CD8⁺ anti-MUM3 T cells were isolated from the blood of melanoma patient LB33[41]. Briefly, PBMCs were isolated from the patient's blood by Lymphoprep (Axis-Shield; #1114547) and stained with HLA-A68 tetramers loaded with MUM3 peptide. A semiclonal population was sorted using a FACS AriaIII. CD8⁺ anti-MUM3 T cells were amplified by bi-monthly stimulation using irradiated ($10^2$ Gy) feeder cells and LB33-EBV stimulatory cells pulsed with 10 ng ml⁻¹ of the MUM3 peptide (1 CD8⁺: 1 LB33-EBV MUM3⁺: three feeder cells). T cells were cultured in Iscove Medium complemented with 10% human serum, AAG, 50 U ml⁻¹ hIL-2 (Proleukin, Novartis) and 10 ng ml⁻¹ hIL-7 (R&D Systems; #207-IL). Frozen CD8⁺ anti-MUM3 cells were thawed, stimulated and cultured for 6 days before being injected in the mice.

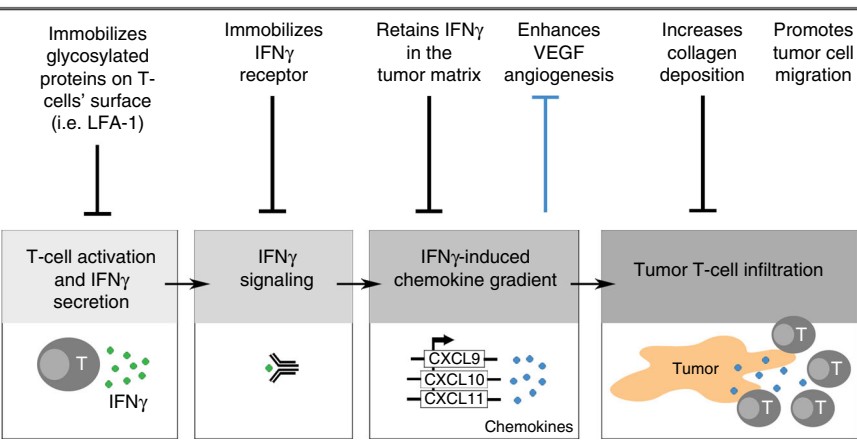

**Fig. 9** Scheme summarizing the main protumoral effects known for galectin-3 in the tumor microenvironment. Galectin-3 has been described to immobilize glycosylated proteins on the surface of T cells[28, 30, 31, 32, 33], IFNγ receptor on human fibroblasts and B cells[53], and VEGF receptor in human endothelial cells[27]. In addition, galectin-3 favors collagen deposition by macrophages[48] and tumor cell migration[54]. This figure does not intend to make an exhaustive review about all the effects published for galectin-3 but to highlight the ones that, in our opinion, can be working simultaneously in the tumor microenvironment

*Galectin antagonists.* N-Acetyl-D-Lactosamine (LacNAc; GLY008) and Tetra-N-Acetyl-D-Lactosamine (TetraLacNAc; GLY007-3) were obtained from Elicityl; GM-CT-01 was kindly provided by Galectin Therapeutics; anti-human Galectin-3 mAb Rat IgG2a was obtained from MabTech AB (#3720-3) and anti-human Galectin-3 mAb clone B2C10 mouse IgG1 from BD Pharmingen (#556904). Control isotype antibodies were a polyclonal rat IgG (Bioxcell; BE0094) and a home-made mouse monoclonal IgG1 (B8401H5).

*Cytokines.* Recombinant human glycosylated IFNγ was produced by CHO cells (SinoBiological Inc.; #11725-HNAS) or by HEK cells (BioVision; #7271). Its unglycosylated counterpart was produced by *E. coli*, (ThermoScientific; PHC4031). The glycosylated IFNγ from SinoBiological was used for all the experiments except specified otherwise. Purified recombinant human glycosylated IL-12 (Prospec; cyt-101) was produced by HEK cells. Purified recombinant human glycosylated CCL5 (Origene; TP303799) was produced by HEK cells. Professor J. Van Snick (Ludwig Institute for Cancer Research, Brussels) kindly provided purified recombinant human glycosylated CCL1. Collagenase D from *Clostridium Histolyticum* was obtained from Sigma-Aldrich (#11088858001). AMG478 was purchased from Tocris (#4487).

**Galectin-coated beads and in vitro bioassays.** Dynabeads M-280 Streptavidin (Invitrogen; #11-205-D) were incubated overnight at 4 °C with saturating amounts of recombinant human galectin-3 biotinylated in the C-terminal cysteine (cloned as described in Petit et al.[30]). Then, beads were washed three times with PBS 2 mM EDTA, 0.05 % Tween-20. Different amounts of galectin-3-coated beads were incubated with the cytokines for 1 h at room temperature in a shaker. After magnetic sorting of the beads, the cytokines present in the supernatants were measured by ELISA (for IFNγ- Life Technologies sandwich antibodies: clone 350B 10G6 #AHC4432 and 67F 12A8 #AHC4539; for IL-12- eBioscience #88-7126), or by Bioplex (Biorad). For CCL1 and 5 see Supplementary Methods.

**Cytokine diffusion in vitro assay.** Collagen-I matrices were prepared as follows: for 1 ml final volume, 100 μl Minimum Essential Medium 10× (Gibco; #11-430-030), 4 mg collagen-I (Nutragen Advanced BioMatrix; #5010-D), 100 μl of 7.5% sodium bicarbonate and distilled water. The mix was distributed on P24 Transwell inserts (Costar 0.4 μm pore diameter, CLS-3412-24EA), 100 μl per insert. Matrix polymerization took 40 min at 37 °C. Then, 100 μl per insert of Iscove Medium with or without 1 μg recombinant galectin-3 was added and incubated 30 min at 37 °C. The unbound galectin-3 was aspirated and washed with 100 μl medium. Transwell inserts with the collagen ± galectin-3 were placed on P24 wells with 600 μl per well of medium. Glycosylated IFNγ or IL-12 (25 or 7 ng per insert, respectively) was added on top of the collagen matrix with or without 5 mM LacNAc in 100 μl medium. Plates were placed for 4 h at 37 °C. Then, Transwell inserts were removed and the cytokines in the medium were measured by ELISA (described above).

**CXCL9 mRNA induction in vitro.** Tumor cells were seeded in P6 wells. Three days later, confluent cells were treated for 4 h with 50 ng ml⁻¹ IFNγ, 5 mM LacNAc, 5 mM sucrose, and/or 5 μg ml⁻¹ neutralizing anti-human IFNγR1 mouse IgG1 antibody (GIR-208 clone; R&D MAB6732). Cells were lysed and RNA was extracted as explained below.

**RNA extraction and reverse transcription and real time-PCR.** For the in vitro experiments, cells from confluent P6 wells were lysed in the plate and RNA was extracted using the Invitrap Spin Cell RNA kit (Stratec Molecular). For tissue samples, RNA was extracted using Tripure Isolation Reagent (Roche; #11667165001). Reverse transcription was performed with 2 μg of total RNA with PrimeScript RT Reagent Kit (Takara; RR037A) and no random 6 mers were added. Quantitative PCR amplifications were performed using Premix Ex Taq kit (Takara; RR390W), 300 nM of each primer, 150 nM of the probe, 1.25 μl of the cDNA in a final reaction volume of 20 μl. RT-PCR were run on an ABI StepOnePlus Sequence Detector (Applied Biosystems): 95 °C for 30 s, 40 cycles of 95 °C for 3 s and 60 °C for 30 s. Reactions were performed in technical triplicates. More details and primers used can be found in Supplementary Methods and Supplementary Table 1.

**Mice experiments.** NOD.Cg-Prkdc^scid^Il2rg^tm1Wjl^/SzJ (NSG) mice were from Jackson Laboratory. Mice were bred at the animal facility of the Université catholique de Louvain, Belgium. Handling of mice and experimental procedures were conducted in accordance with national and institutional guidelines for animal care. NSG mice of both sexes and 8–10 weeks old were sub-lethally irradiated (1.22 Gy) the day before tumor subcutaneous injection to help tumor engraftment. Antibiotics (Baytril 0.01%; Bayer) were added the day before irradiation to the drinking water and changed every 3–4 days. Tumor cells, 5 millions per mouse, were injected in a medium-Matrigel (Corning #354234) mix (1:1) and tumor growing was monitored twice per week. LB33-MEL tumors reached a size of 500 mm³ (volume calculated with the ellipsoid formula LxWxH/2) within about one month. SKBR3-derived cell lines grow with difficulty in NSG mice, producing tumors with low incidence and reaching only small sizes (10–50 mm³).

For ex vivo treatments, mice were sacrificed once the tumors reached a volume of about 500 mm³ (LB33-MEL tumors), or after 40 days (SKBR3-derived tumors). Tumors were cut in pieces and each piece was placed in a P48 well with the corresponding treatment, in 500 μl medium, and incubated for 6–8 h in a cell incubator. Pieces were washed with cold PBS, frozen in liquid nitrogen, and stored at −80 °C until RNA extraction.

For the in vivo experiments, once the tumors reached 500 mm³, mice were anesthetized with Domitor (Orion Pharma) and Anesketin (Eurovet) mix. Tumors were injected using a Hamilton syringe (1702- Needle gauge 27 point 4) with 10 μl of each treatment in PBS 0.2% Trypan blue in the center of the tumor. Other colorants were discarded as they were affecting IFNγ-induced CXCL9 expression (Supplementary Fig. 5b). Mice were woken up with Antisedan (Orion Pharma). The day after, mice were sacrificed and tumors were collected and frozen in OCT media (Electron Microscopy; #62550) using dry ice. Samples were stored at −80 °C. Cryostat sectioning was performed along the whole tumor: RNA was extracted from each pool of 30 slices of 20 μm thickness in a tube containing 1 ml of tripure isolation reagent. Between each section, six slices of 20 μm thickness were discarded or kept for immunostaining. Thus, every section represents 0.66 mm of tissue (30 × 20 μm + 3 × 20 μm = 660 μm = 0.66 mm). CXCL9 gradient threshold was referred to the number of CXCL9 mRNA copies per section. To do so, we extrapolated the number of cells per section using the mRNA values obtained from

the reference gene HPRT-1. The copies of CXCL9 mRNA were calculated by doing a standard curve with a plasmid coding for hCXCL9.

For the infiltration experiments, once the tumors reached 500 mm$^3$, treatments were injected in the center of the tumor as described above and, 2 days later, $10^7$ CD8$^+$ anti-MUM3 T cells (day 6 after stimulation) were loaded with 1.5 μM CMFDA (Life Technologies) and injected through the tail vein. Four days after T-cell injection, mice were anesthetized, and blood was collected by retro-orbital injection. Mice were sacrificed immediately after, and the tumor and the spleen were collected. Tumors were homogenized using gentleMACs Dissociator (3 x program h_tumor_01; Miltenyi Biotec) in an enzymatic mix with 100 U ml$^{-1}$ collagenase-I (Roche; #17100-017), 100 U ml$^{-1}$ collagenase-II (Roche; #17101-015), and 1 U ml$^{-1}$ dispase (Roche; #17105-041), following the manufacturer's instructions. Spleens were homogenized by mechanical disruption in cold PBS 2 mM EDTA 2 % human serum. Blood was centrifugated 10 min at 2000×g at 4 °C. Blood pellets and spleen homogenates were resuspended in 155 mM ammonium chloride, 10 mM potassium bicarbonate 0.1 mM EDTA to lyse the erythrocytes. Cells from blood, spleen, and tumor homogenates were then stained for flow cytometry analysis. For CXCR3 blocking experiments, we followed the protocol published by Cambien et al.[42]. Briefly, T cells were pretreated the day before injection with AMG478 or DMSO at 1 μM. Starting the day of T-cell injection, mice were treated twice daily for four days with peritumoral subcutaneous injections of 5 mg kg$^{-1}$ AMG or DMSO dissolved in 20% 2-hydroxypropyl-β-cyclodextrin (Sigma-Aldrich, # C0926).

To assess the tumor growth, mice were treated 15 days after tumor implantation. Treatments were injected in the center of the tumor as described. Treatments consisted on IFNγ (10 ng per tumor) and either a control ratIgG antibody (100 ng per tumor) or galectin-3 antagonists (0.1 μmol per tumor LacNAc and 100 ng per tumor anti-galectin-3 ratIgG). The same day, each mouse received through the tail vein $10^7$ CD8$^+$ anti-MUM3 autologous T cells at day 6 after stimulation and intraperitoneally 200 ng of hIL-2 (Sigma-Aldrich; Proleukin). Tumor growth was measured with a caliper every 3/4 days from the day of treatment. Tumor volume was calculated as formerly explained. Mice received three more intraperitoneal injections ("boosts"), one every seven days after treatment, consisting on hIL-2 (200 ng per mouse) and either control ratIgG antibody (50 μg per mouse) or galectin-3 antagonists (40 μmol per mouse LacNAc and 50 μg per mouse anti-galectin-3 ratIgG). Mice were sacrificed when the tumor reached the end point volume of 1500 mm$^3$. For the experiment shown in Fig. 8c, 200 ng per tumor of anti-galectin-3 antibody was injected intratumorally.

**Flow cytometry**. Extracellular staining was performed in home-made Hanks' Balanced Salt solution (HBSS) 1% human serum for 30 min at 4 °C on non-fixed cells and in the presence of human and mouse FcR blocking reagents (MACS Miltenyi Biotech; #553141 and #130-059-901). Primary antibodies were incubated for 30 min. Cells were washed and incubated with secondary antibodies when needed. Samples were kept at 4 °C in the dark until acquisition by a FACS Fortessa (BD). Data were analyzed using FlowJo software. Intracellular staining was performed at room temperature on fixed cells (5 min with 4% para-formaldehyde; Merck #4005). Cells were permeabilized with HBSS 1% human serum 0.1% saponin (Sigma; S-1252). Staining with primary antibodies directly coupled to fluorochromes (used at 5-1 μg ml$^{-1}$) and the washing steps were performed in the permeabilizing buffer. Primary antibodies used for flow cytometry were purchased from Biolegend: anti-human/mouse granzyme B Alexa-647 mouse IgG1 (#515405), anti-human CD183 (CXCR3) BV-510 mouse IgG1 (G025H7 clone; #353725), anti-human/mouse galectin-3 alexa-647 ratIgG2a (#125416), anti-human CD279 (PD-1) PE-Cy7 mouse IgG1 (E12.2H7 clone; #329917), anti-human CD8a Alexa Fluor 647 mouse IgG1 (clone HIT-8a; #300918); anti-human CD137 PE mouse IgG1 (clone 4B4-1; #309804). Anti-human PD-L1 PE (MIH-1 clone, #557924), anti-human CD8 PE (SK-1 clone; #4115585) and anti-human IFNγ PE mouse IgG1 (B27 clone; #554701) were obtained from BD Pharmingen. Control isotype antibodies were: mouse IgG1-PE-Cy7 (MOPC-21 clone; BD Pharmingen #557872); mouse IgG1-BV510 (MOPC-21 clone; Biolegend #400171); mouse IgG1-PE (BD Bioscience #345816); mouse IgG1-Alexa Fluor 647 (MOPC-21 clone; Biolegend #400130); and rat IgG2a BV421 (RTK2758 clone; Biolegend #400535). Sphero-rainbow beads were added as a counting reference (BD Bioscience #556288).

**Immunohistochemistry**. Tissue slices of 8 μm were fixed with 10% formaldehyde for 5 min, washed with TBS for 3 min and permeabilized with TBS 0.05% Tween-20 for 3 min. Endogenous enzymes were blocked 15 min with Dako dual endogenous enzyme blocker (Dako; S2003). Tissue unspecific binding was avoided by 1 h incubation with 5% bovine serum albumin, 2% powder milk, 1% human IgG in TBS 0.05% Tween-20. Primary antibody, anti-human CXCL9 goat pAb was used at 1 μg ml$^{-1}$ (R&D; AF392), diluted in REAL antibody diluent (Dako; S2022) and incubated for 1 h. After three washes in TBS 0.05 % Tween-20, the secondary antibody ImmPress anti-goat-polyHRP (Vector; MP-7405) was incubated for 45 min. Slides were washed again three times and then, the substrate AEC + chromogen (Dako; K4005) was incubated for 15 min. Reaction was stopped by immersion in demineralized water. Counterstaining with hematoxylin (solution according to Mayer; Sigma-Aldrich #51275) was done previous to mounting the slide with Faramount Aqueous Mounting Media (Dako; S3025). Images were acquired using the Mirax Digital Slide Sytem (Zeiss) and analyzed with Image J software.

**Treatment of human tumor biopsies**. Biopsies from human tumors were provided by the Cancer Centre Biobank of Cliniques universitaires Saint-Luc with the informed consent of each patient (Brussels, Belgium, project CDCUCLR). Tumors were harvested after surgery and cut in pieces. Each piece was immediately treated in P48 wells with 0.5 ml medium for 6–8 h in the cell incubator. Treatments consists on 50 ng ml$^{-1}$ of IFNγ, 10 mM Sucrose, 10 mM LacNAc, 200 μg ml$^{-1}$ GM-CT-01, 30 μM TetraLacNAc, 10 μg ml$^{-1}$ of anti-galectin-3 antibodies, and 5 μg ml$^{-1}$ of the neutralizing anti-IFNγR1 antibody. After treatment, each piece was washed with cold PBS, frozen in liquid nitrogen, and stored at −80 °C.

**Statistic analysis**. Statistic analysis was performed using JMP Pro 12 software and the statistical method used is specified in the figure legends for each experimental set. Generally, Kruskal–Wallis test (nonparametric analysis of variance (ANOVA)) was used for in vitro experiments and Wilcoxon matched-pairs signed rank test was used for ex vivo and in vivo experiments. Dunn's Multiple Comparison Correction Test was added when more than two groups were compared. All tests were two-sided and variance between groups was similar. Tumor growth experiments were analyzed by two-way ANOVA test. For animal studies sample size was estimated previously using T-test or ANOVA. Animals of both sexes were used randomly and data was collected in blinded fashion.

**Data availability**. Data supporting the findings of this study are available within the article and its Supplementary Information Files or from the corresponding author upon reasonable request.

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

## Acknowledgements

We thank Didier Colau, Jacques Van Snick, Emma Salomonsson, Ilse Gutierrez, and Pierre Coulie for reagents; and Emmanuel Donnadieu, Alain Trautmann, Nadege Bercovici, Pierre Coulie, Francisco Sanchez-Madrid, and Nicolas van Baren for manuscript revision. This work was supported by grant #2010-175 from the Fondation contre le Cancer (Belgium), grant #3.4514.12 from the Fonds de la Recherche Scientifique Médicale-FRSM (Belgium). M.G.-A. is supported by grant 14-0189 from Worldwide Cancer Research.

## Author contributions

M.G.-A. conceived and designed the experiments, performed and supervised experiments, analyzed the data, wrote the manuscript, and secured funding. C.W. and T.H. performed experiments. P.v.d.B. revised the manuscript and provided reagents, supervised experiments, and provided funding.

## Additional information

**Competing interests:** The authors declare no competing financial interests.

