## [Peer Review File · Nature Communications]

Reviewers' comments:

Reviewer #1 (Remarks to the Author):

The manuscript by Monica Gordon-Alonso and colleagues is an interesting study of tumor immunology. According to the formulated hypothesis, tumor-expressed galectin-3 scavenges glycosylated interferon-gamma (IFN-gamma) in such a way that it does not induce CXCL9/10/11 chemokines anymore, thus leading to less infiltration of T lymphocytes attacking tumors and resulting in higher tumor burden and poorer prognosis of patients. This is an interesting glycobiological study about galectin-sugar interactions. It is rather strange that, whereas galectin-3 is the key molecule of this study, galectin-3 is mentioned in the additional but not in the main title. The manuscript contains a large amount of primary data and is well written. The hypothesis is novel but very challenging to substantiate, because of the multitude of molecules involved, because of the existing redundancies in chemokines and galectins and because it is difficult to discriminate between direct and indirect effects with the used experiments. However, the authors are stimulated to resolve a number of theoretical and practical issues.

Major Comments

1. Immunological effects of sugars and lectins are complex because these molecules have many functions, because of heterogeneity of sugars and because of an absence of selectivity of many reagents used in this study. Therefore, it is a real challenge to attribute specific effects to the oligosaccharides and therefore many controls are necessary. It is advised to address these aspects in the Introduction section. Now, the reader is given a limited insight into the biology with an oversimplification of known facts.
2. T cell recruitment to solid tumors is regulated by more chemokines than the mentioned CXCL9/10/11 and it is surprising that the effects would be mainly attributable to one single member of chemokines. In addition, galectins come as a family in tumor biology with more functions than just liganding glycoproteins. Why would galectin-3 be the major/only critical factor? The used sugar antagonists are certainly not specific for galectin-3. In addition, is IFN-gamma the major molecule that is antagonized? This is probably not the case and this needs to be clarified here.
3. The used interferon-gamma is from hamster cells (or HEK cells) and thus contains hamster (or human type) oligosaccharides. Is the complete cytokine preparation blocked by the used sugars? Probably not, because the preparation contains a mixture of glycoforms. What is the fraction that is not blocked? Which fraction is antagonized? A similar remark is made about the used glycosylated chemokines. The reader needs to understand this problem. Therefore, it is advised to provide a table with the used glycoproteins, with the structural information about the attached oligosaccharides and the interactions of these sugars with galectin-3.
4. One relevant control of the used system is the comparison between glycosylated IFN-gamma and the aglycosyl cytokines (e.g. expressed in *E. coli*). Whereas this type of control is used for in vitro experiments (Fig 1a), it was not used in the tumor growth experiments. These controls need to be provided. According to the formulated hypothesis, aglycosyl IFN-gamma should work better against the tumor, because it will be 100% available. If the opposite will be observed, alternative explanations need to be found.
5. Another practical question (and extra control experiments) relates to the collagenase D experiments. Please provide evidence that collagenase D cleaves galectin-3 and not IFN-gamma. It is furthermore relevant to compare aglycosyl with glycosylated IFN-gamma, because it is known from seminal literature data that oligosaccharides (in particular large N-linked structures) protect glycoproteins against proteolytic attacks.

Minor Comments

1. According to the international nomenclature rules, interferons are written with hyphens: interferon-gamma (IFN-gamma)
2. Page 2, line 2: infiltrated (instead of infiltrate)

3. Page 3, line 10: these (instead of them)

Reviewer #2 (Remarks to the Author):

This is an interesting, well-designed and well-written study that proposes a novel mechanism by which galectin-3 in the tumor matrix retards IFN γ diffusion and thus Teff recruitment to promote immune evasion by the tumor. The manuscript is from a group that has published important work on the role of galectin-3 in tumor immune evasion and the current work builds on their previously published papers. As tumor immune evasion is a critically important topic both intellectually and clinically, novel insights into this complex process are timely and important. While the concept of extracellular galectins binding cell surface and secreted glycoproteins to organize membrane domains and influence cell signaling is well-established, the current findings that galectin-3 binds IFN γ and that galectin-3 antagonists increase IFN γ diffusion as well as the in vivo response to the tumor are original and well-supported.

There are a few specific areas that could be improved or alternative interpretations considered.

In the in vivo experiments, the authors inject galectin-3 antagonists intratumorally, and much of their prior work has examined the effects of galectin-3 antagonists in vitro or delivered locally in vivo. It would be helpful, perhaps in the Discussion, to discuss their approach in relation to previous studies by others that delivered a galectin-3 antagonist orally or intravenously, as the requirements for route of administration would be critical to consider when one evaluates the translational potential of the findings.

The galectin-glycoprotein lattices that exist on cells, between cells, and between cells and matrix are complex and probably interact and overlap in a number of ways. It is understandable that the authors, for the sake of clarity, would focus on the novel finding that galectin-3 can bind IFN γ and thus describe all the downstream effects as resulting from retarding the diffusion of IFN γ . However, as this group published last year in Nat Commun, galectins bind several cell surface glycoproteins on CD8 cells. Thus, the authors should at least acknowledge that additional effects, e.g. the availability or lateral mobility of the hIFN γ receptor, could also be affected by galectin-3 binding and contribute to the immune suppressive effect. Especially for those new to the galectin field, one would not want to give the impression that releasing IFN γ would be the only effect of a galectin-3 antagonist. Perhaps a schematic in the Discussion would be helpful.

Linda G. Baum

Reviewer #3 (Remarks to the Author):

Gordon-Alonso and colleagues put forward interesting data regarding tumor-derived galectin 3, the formation of matrices that capture glycosylated IFN γ , and how the capturing of IFN γ adversely affect the production of chemo-attractants and infiltration of CD8 T cells. This report is well communicated, data are novel to the field of immune-oncology, and potentially relevant by providing another angle to enhance the number of TILs and treat tumors. I do have suggestions with respect to methodology and discussion.

Major suggestions:

- The NSG experiments lack necessary controls. First, intratumoral addition of IFN γ in a setting where LB33-specific CD8 T cells were administered lowers the translational value of these experiments. I do understand the rationale (provision of IFN γ that becomes trapped), but in a patient setting available IFN γ will not be administered intratumorally, and has to come from endogenous or adoptively transferred T cells. Authors should therefore include adoptive T cell

transfer experiments without pre-treating tumors with IFN γ . Second, it is recommended to use LB33 in which the CXCL9 gene is knocked out to assess whether CXCL9 is truly required for enhanced T cell infiltration. Alternative options include neutralization of CXCL9 with antibodies or use T cells in which CXCR3 has been knocked out.

- In extension to the above comment, and to investigate mechanisms, authors should consider multiplex analysis of cytokines and chemoattractants of medium that is conditioned by ex vivo xenografts following in vivo treatments.
- To further strengthen the role of CXCL9, authors should also look for (or when existing, refer to) correlations in patient tumor samples, either high or low with respect to CD8 T cells, between galectin 3 and CXCL9 expressions. To translate findings to a clinical setting, authors should discuss how they would propose to stratify patients, and how to administer galectin antagonists locally.

Minor suggestions:

- Fig 3d, e: also indicate absolute CXCL9 mRNA expression (not relative to IFN γ treatment), and confirm that these levels (following IFN γ treatment) are higher with galectin mutant. In addition, authors should comment why in Fig 3e, one observes no reduction in CXCL9 mRNA with α -IFN γ R1.
- Fig 4 is redundant (data already present in Table 1).
- Fig 5b: to better understand the contribution of lattices covering tumor cells versus those that cover collagens, it would be helpful to add the fraction of tumor content (based on HE or a tumor marker) along the tumor sections. It would also help to put figs 5 and 6 into a single figure.
- Fig 7b: please represent data for tumor only (not vs spleen); and Fig 7d: please represent % and MFI separately.
- Fig 8: NaLac and α -Gal3 treatments are combined, whereas these treatments are given separately in Fig 7. Please also add separate treatments in Fig 8. In addition, authors should not refer to these data as tumor regression, yet speak of delay in tumor outgrowth.
- Discussion: please add what fraction of IFN γ normally is N-glycosylated. Also, comment on the lack of effect of galectin-3 antagonists towards chemoattractants (Fig 1d, no effect of lactose; and Fig S1); possibly unexpected since chemoattractants are glycosylated.
- Discussion: line 361, first time 'galectin-3' should be 'galectin-3 antagonists'.

A set of new experiments and analysis has been performed replying to each of the reviewers concerns. New data address the role of IFN γ glycosylation in vivo (Reviewer#1) and that of CXCL9 and other cytokines in the observed enhanced T cell infiltration (Reviewer#1 and Reviewer#3).

We thank the reviewers for their comments. We strongly feel that they helped improving our study. The questions/comments of reviewers are written in black, our answers are written in blue. The modifications performed in the manuscript are indicated in blue in the main text, and as New Figures. Additional data, which are not included in the revised manuscript, are shown in annexes for the reviewers. If one reviewer considers that some of these additional data are crucial for the readers, we are ready to include them as supplementary figures.

Reviewer #1:

The manuscript by Monica Gordon-Alonso and colleagues is an interesting study of tumor immunology. According to the formulated hypothesis, tumor-expressed galectin-3 scavenges glycosylated interferon-gamma (IFN-gamma) in such a way that it does not induce CXCL9/10/11 chemokines anymore, thus leading to less infiltration of T lymphocytes attacking tumors and resulting in higher tumor burden and poorer prognosis of patients. This is an interesting glycobiological study about galectin-sugar interactions.

It is rather strange that, whereas galectin-3 is the key molecule of this study, galectin-3 is mentioned in the additional but not in the main title.

We have changed the title accordingly and included “galectin-3”. The new title is “Galectin-3 captures interferon-gamma in the tumor matrix, reducing chemokine gradient production and T-cell tumor infiltration”

The manuscript contains a large amount of primary data and is well written. The hypothesis is novel but very challenging to substantiate, because of the multitude of molecules involved, because of the existing redundancies in chemokines and galectins and because it is difficult to discriminate between direct and indirect effects with the used experiments. However, the authors are stimulated to resolve a number of theoretical and practical issues.

Major Comments

1. Immunological effects of sugars and lectins are complex because these molecules have many functions, because of heterogeneity of sugars and because of an absence of selectivity of many reagents used in this study. Therefore, it is a real challenge to attribute specific effects to the oligosaccharides and therefore many controls are necessary. It is advised to address these aspects in the Introduction section. Now, the reader is given a limited insight into the biology with an oversimplification of known facts.

Following the reviewer suggestion, we have modified the introduction (page 3, paragraphs 4-5) to alert the readers about the heterogeneity of sugars and the pleiotropy of galectins, hence the difficulty to analyze their specific effects.

2. T cell recruitment to solid tumors is regulated by more chemokines than the mentioned CXCL9/10/11 and it is surprising that the effects would be mainly attributable to one single member of chemokines.

We agree with the reviewer about the complexity of the different actors and it is why we have performed *in vivo* experiments in NSG mice, where the T cells are most probably influenced only by human chemokines produced by the human tumor cells.

We are also aware that T cell infiltration in tumors can be mediated by several chemokines, as mentioned in the introduction (second paragraph). We focused our analyses on CXCL9/10 mainly because these chemokines are specifically induced by IFN γ . We have now performed a new experiment (New Figure 7d and Supplementary Fig. 9a), where we have examined which chemokines were responsible for the increased T cell infiltration. We know now that the increase in T cell infiltration observed when tumors were injected with IFN γ and anti-galectin-3 antibody is abolished when mice are also injected with a CXCR3-specific chemical inhibitor, AMG478 (Cambien *et al.* 2009 British Journal of Cancer) (New Figure 7d). This indicates that indeed CXCL9, CXCL10, and/or CXCL11 are the main chemokines involved in the enhanced T cell infiltration induced by the anti-galectin-3 antibody.

In addition, galectins come as a family in tumor biology with more functions than just liganding glycoproteins. Why would galectin-3 be the major/only critical factor? The used sugar antagonists are certainly not specific for galectin-3.

We agree that LacNAc is not a specific antagonist for galectin-3, but we have also used a monoclonal specific anti-galectin-3 antibody (Figure 7b). Treatment with the anti-galectin-3 antibody also increases T cell infiltration similarly to LacNAc, and delays tumor growth (New Figure 8c). Although we cannot exclude that other galectins or lectins may be involved in cytokine retention, inhibiting galectin-3 is sufficient to boost IFN γ -induced chemokine production (Discussion, last two sentences of the 4th paragraph).

In addition, is IFN-gamma the major molecule that is antagonized? This is probably not the case and this needs to be clarified here.

We share the reviewer's feeling that galectin/cytokine/matrix interactions can have a broader effect involving several cytokines. This is exactly why we performed our experiments in immunocompromised NSG mice where human IFN γ is the only cytokine that can induce human CXCL9 production, allowing us to specifically evaluate the role of IFN γ by measuring CXCL9 expression. This specificity is proven by our *in vitro* and *ex vivo* data in human biopsies and xenografts performed in the presence of an anti-IFN γ R-1 antibody where CXCL9 induction was

entirely dependent on IFN γ signaling (Figure 3b, 3d and New Supplementary Figure 4b). We have added a clause in the first paragraph of the discussion in the revised version to better clarify this point: “Our preliminary data show interaction of galectin-3 with IL-12 as well, but further studies are needed to test the effect of various galectins on other glycosylated cytokines, and more importantly, to assess their biological significance.”

3. The used interferon-gamma is from hamster cells (or HEK cells) and thus contains hamster (or human type) oligosaccharides. Is the complete cytokine preparation blocked by the used sugars? Probably not, because the preparation contains a mixture of glycoforms. What is the fraction that is not blocked? Which fraction is antagonized? A similar remark is made about the used glycosylated chemokines. The reader needs to understand this problem. Therefore, it is advised to provide a table with the used glycoproteins, with the structural information about the attached oligosaccharides and the interactions of these sugars with galectin-3.

Most of the glycosylated IFN γ was captured by galectin-3-beads in the *in vitro* experiments, as less than 10% of the total IFN γ was free after 1 h incubation with galectin-3 beads at a ratio galectin/IFN γ of 10 (Figure 1A; two additional independent experiments are shown in Annex 1a for the reviewer).

To provide further information about the fraction of IFN γ that is glycosylated, we have now performed a Coomassie staining for glycosylated (produced in CHO) and unglycosylated (produced in *E.coli*) IFN γ samples. The glycosylated IFN γ gives two bands at 25 KD and 21 KD, and the unglycosylated IFN γ gives one band at 17 KD (Annex 1b). These data show that the IFN γ produced in CHO cells is glycosylated as both bands are clearly higher than the one observed for unglycosylated IFN γ . For glycosylated CCL5 we also observed a unique band at 15 KD (unglycosylated CCL5 runs at 10KD) (data not shown).

As suggested by the reviewer, we have added in the revised version a table with the glyco-structural information of the cytokines and chemokines tested and their binding to galectin-3 (New Supplementary figure 1d).

4. One relevant control of the used system is the comparison between glycosylated IFN-gamma and the aglycosyl cytokines (e.g. expressed in *E. coli*). Whereas this type of control is used for *in vitro* experiments (Fig 1a), it was not used in the tumor growth experiments. These controls need to be provided. According to the formulated hypothesis, aglycosyl IFN-gamma should work better against the tumor, because it will be 100% available. If the opposite will be observed, alternative explanations need to found.

Unfortunately, things are more complicated than they seem. Glycosylated and unglycosylated IFN γ cannot be directly compared in *in vivo* experiments, because their activities are different in both potency and half-life. *In vitro*, incubations are short, and unglycosylated IFN γ was more potent than glycosylated IFN γ (see Annex 2). However, *in vivo*, the effects of unglycosylated/glycosylated IFN γ are difficult to predict due to the shorter half-life time of the unglycosylated IFN γ (Opdenakker et al. 1995 FASEB J.). Therefore, the reviewer’s claim “aglycosyl IFN-gamma should work better against the tumor, because it will be 100% available” does not take into account the differences in both potency and half-life of the two IFN γ .

We have nevertheless performed a new *in vivo* experiment and analyzed tumor growth in mice treated with unglycosylated IFN γ with or without anti-galectin-3 antibody. The amount of unglycosylated IFN γ injected was chosen to be equivalent to the amount of glycosylated IFN γ inducing *in vitro* a high production of CXCL9. We did not observe a delay in tumor growth when

unglycosylated IFN γ was injected independently of anti-galectin-3 antibodies addition (Annex 3). Our tentative conclusion is that the unglycosylated IFN γ has a very short half-life *in vivo* and it is not able to induce an effective signal.

5. Another practical question (and extra control experiments) relates to the collagenase D experiments. Please provide evidences that collagenase D cleaves galectin-3 and not IFN-gamma. It is furthermore relevant to compare aglycosyl with glycosylated IFN-gamma, because it is known from seminal literature data that oligosaccharides (in particular large N-linked structures) protect glycoproteins against proteolytic attacks.

Collagenase D cleaves at the consensus aminoacid sequence P-X-G-P. This motif is found in collagen (64 motives/molecule for collagen alpha-I) and galectin-3 (2 motives/molecule) but not in IFN γ .

We have now evaluated experimentally whether collagenase D cleaves collagen-I, galectin-3, and IFN γ and also the influence of IFN γ glycosylation. Collagenase D was able to cleave both collagen-I and galectin-3 as observed by Coomassie staining (Annex 4b). Glycosylated IFN γ was not cleaved by collagenase D whereas unglycosylated IFN γ was cleaved. This can be explained because the collagenase D commercially available is contaminated with trypsin and other proteins (observed the many bands obtained in the lane with collagenase D alone, last lane of the gel). Since Coomassie stainings are performed in the absence of serum (the various serum proteins would hide the proteins in study) trypsin traces may cleave IFN γ . Glycosylated IFN γ seems more resistant to trypsin and other proteases cleavage which is coherent with its longer half-life *in vivo*.

In another set of experiments, LB33-MEL cells were incubated with IFN γ glycosylated or not in the presence of collagenase D. Noteworthy, for these *in vitro* experiments, cells were cultured in complete media allowing the serum to block trypsin traces. As shown in Annex 4a, IFN γ -induced CXCL9 production was not affected by the presence of collagenase D. This was true whether IFN γ was glycosylated or not.

Minor Comments

1. According to the international nomenclature rules, interferons are written with hyphens: interferon-gamma (IFN-gamma)

We have corrected the mistake.

2. Page 2, line 2: infiltrated (instead of infiltrate)

We have corrected the mistake.

3. Page 3, line 10: these (instead of them)

We have corrected the mistake.

Reviewer #2 :

This is an interesting, well-designed and well-written study that proposes a novel mechanism by which galectin-3 in the tumor matrix retards IFN γ diffusion and thus T_H1 recruitment to promote immune evasion by the tumor. The manuscript is from a group that has published important work on the role of galectin-3 in tumor immune evasion and the current work builds on their previously published papers. As tumor immune evasion is a critically important topic both intellectually and clinically, novel insights into this complex process are timely and important. While the concept of extracellular galectins binding cell surface and secreted glycoproteins to organize membrane domains and influence cell signaling is well-established, the current findings that galectin-3 binds IFN γ and that galectin-3 antagonists increase IFN γ diffusion as well as the *in vivo* response to the tumor are original and well-supported.

There are a few specific areas that could be improved or alternative interpretations considered.

In the *in vivo* experiments, the authors inject galectin-3 antagonists intratumorally, and much of their prior work has examined the effects of galectin-3 antagonists *in vitro* or delivered locally *in vivo*. It would be helpful, perhaps in the Discussion, to discuss their approach in relation to previous studies by others that delivered a galectin-3 antagonist orally or intravenously, as the requirements for route of administration would be critical to consider when one evaluates the translational potential of the findings.

We designed our experiments to provide the following proof of concept: a glycosylated cytokine can be retained at the extracellular matrix by galectins. We completely agree with the reviewer that the translational potential of the findings is yet to be proven. As suggested by the reviewer, we have included a new paragraph in the discussion of the revised version: "This study brings the new concept of tumor-secreted lectins reducing the availability of glycosylated soluble factors in the tumor microenvironment. Further studies are clearly required to test the translational potential of these findings. Our observations suggest that local galectin inhibition during cytokine secretion is needed to boost the cytokine function. We have already observed that some human tumors with high T cell infiltration contained galectin-retained IFN γ that was effectively released by short *ex vivo* treatment with galectin antagonists. The effect of longer treatments and systemic administration of galectin antagonists for liberating galectin-retained cytokines has yet to be proven."

The galectin-glycoprotein lattices that exist on cells, between cells, and between cells and matrix are complex and probably interact and overlap in a number of ways. It is understandable that the authors, for the sake of clarity, would focus on the novel finding that galectin-3 can bind IFN γ and thus describe all the downstream effects as resulting from retarding the diffusion of IFN γ . However, as this group published last year in *Nat Commun*, galectins bind several cell surface glycoproteins on CD8 cells. Thus, the authors should at least acknowledge that additional effects, e.g. the availability or lateral mobility of the hIFN γ receptor, could also be affected by galectin-3 binding and contribute to the immune suppressive effect. Especially for those new to the galectin field, one would not want to give the impression that releasing IFN γ would be the only effect of a galectin-3 antagonist. Perhaps a schematic in the Discussion would be helpful.

Linda G. Baum

Indeed, we tried to facilitate the legibility of the picture. We thank the reviewer for his suggestion and have included in the revised manuscript a graphical abstract about the multiple galectin-3 effects in the tumor (New Figure 9).

Reviewer #3:

Gordon-Alonso and colleagues put forward interesting data regarding tumor-derived galectin 3, the formation of matrices that capture glycosylated IFN γ , and how the capturing of IFN γ adversely affect the production of chemo-attractants and infiltration of CD8 T cells. This report is well communicated, data are novel to the field of immune-oncology, and potentially relevant by providing another angle to enhance the number of TILs and treat tumors. I do have suggestions with respect to methodology and discussion.

Major suggestions:

- The NSG experiments lack necessary controls. First, intratumoral addition of IFN γ in a setting where LB33-specific CD8 T cells were administered lowers the translational value of these experiments. I do understand the rationale (provision of IFN γ that becomes trapped), but in a patient setting available IFN γ will not be administered intratumorally, and has to come from endogenous or adoptively transferred T cells. Authors should therefore include adoptive T cell transfer experiments without pre-treating tumors with IFN γ .

The purpose of the study was indeed to test if IFN γ was trapped by galectins and if injecting anti-galectin antibodies releases IFN γ . We never thought nor claim to propose a clinical strategy relying on intratumoral injection of IFN γ . As this study was a proof of concept, we thought it premature to propose a clinical strategy.

However, on the basis of our experiments with human tumor biopsies, treating tumor fragments with LacNAc alone was sufficient to induce an increased CXCL9 expression in half of the tumors. This indicates that endogenous IFN γ was liberated. In addition, in highly infiltrated tumors, CD3 mRNA correlates with the CXCL9 expression induced by LacNAc treatment, suggesting that in these patients, tumor infiltrating T cells had secreted IFN γ . Therefore, in patients with tumors containing activated T cells or NKs it may be sufficient to inject galectin antagonists to boost IFN γ activity.

For the mouse experiments, considering that there is no adaptive and deficient innate immune response, therefore scarce inflammation, we did not expect that T cells could infiltrate efficiently the tumor, produce IFN γ and attract new T lymphocytes.

Nevertheless, to answer to the reviewer request, we have performed a new *in vivo* experiment comparing the tumor growth of tumors pre-treated with anti-galectin-3 antibody with or without IFN γ and then treated with adoptive T cell transfer (New Figure 8c and New supplementary figure 12). As expected, pretreatment with IFN γ is needed to delay the tumor growth, supporting our hypothesis that galectin-3 is hijacking this cytokine in the tumor microenvironment and discarding that anti-galectin-3 antibody is able to reduce the growth rate *per se* by an IFN γ -independent mechanism. Taking together these data and the data previously shown in Figure 8b, where tumor growth is not affected by IFN γ intratumoral administration and adoptively transferred T cells, we can conclude that in our mouse model, the combination of both IFN γ and galectin antagonists are needed for delaying tumor growth.

Second, it is recommended to use LB33 in which the CXCL9 gene is knocked out to assess whether CXCL9 is truly required for enhanced T cell infiltration. Alternative options include neutralization of CXCL9 with antibodies or use T cells in which CXCR3 has been knocked out.

We thank the reviewer for suggesting this additional control. Considering that CXCL9 is not the only chemokine induced by IFN γ , we have decided to block CXCR3, the common receptor for CXCL9/10/11, using the specific chemical inhibitor AMG478 (Cambien *et al.* 2009 British Journal of Cancer). A new *in vivo* experiment has been performed and the results are shown in New

Figure 7d and New Supplementary figure 9. AMG478 treatment abolishes the increased T cell infiltration due to anti-galectin-3 addition.

We concluded that CXCL9/10/11 are the chemokines required for the boosted attraction of T lymphocytes induced by the anti-galectin-3 antibody. We want to remind the reader that we focused in CXCL9 because of its higher specificity for IFN γ signaling. We have also measured CXCL10 induction *in vitro* obtaining similar results (Supplementary Figure 2 and 3).

- In extension to the above comment, and to investigate mechanisms, authors should consider multiplex analysis of cytokines and chemoattractants of medium that is conditioned by ex vivo xenografts following in vivo treatments.

As suggested by the reviewer, a new experiment has been done measuring several cytokines and chemokines secreted *ex vivo* by tumors treated *in vivo*. Tumors were treated *in vivo* with IFN γ with or without anti-galectin-3 antibody, two days later T cells were injected through the tail vein, and four days later tumors were extracted, cut in pieces and incubated *ex vivo* for another 18h. The supernatants were analyzed for several cytokines and chemokines using multiplex technology. The results show no detection of human IL-2, IL-4, IL-6, IL-10, IL-12, GM-CSF and CCL4 (Annex 5). We could observe low amounts of human IL-1beta, IFN γ , TNF α , CCL2 and CCL5 (Annex 5). IFN γ -induced chemokines CXCL9, CXCL10 and CXCL11 were the main chemokines detected, particularly CXCL10 (Annex 5). No statistically significant difference was observed at this time point between the two treatments (Annex 5).

- To further strengthen the role of CXCL9, authors should also look for (or when existing, refer to) correlations in patient tumor samples, either high or low with respect to CD8 T cells, between galectin 3 and CXCL9 expressions.

We thank the reviewer for his suggestion. First, we analyzed a possible correlation between galectin-3 presence and CXCL9 response to galectin antagonists. We measured galectin-3 mRNA in the human biopsies available. Then, tumors were divided in responding or not responding to galectin antagonists. Responding tumors have at least two folds increase in CXCL9 induction after IFN γ +LacNAc regarding IFN γ alone treatment. There was a significant correlation between galectin-3 expression and CXCL9 boosted induction in the responding tumors (New Figure 4a).

Second, we have analyzed the correlation between CD3 content and CXCL9 induction in samples treated with LacNAc alone (without addition of IFN γ). We reasoned that there should be a direct correlation between the internal source of IFN γ (mostly CD3+ lymphocytes) and CXCL9 induction after galectin antagonist treatment. We found indeed a significant correlation between CXCL9 mRNA and CD3 mRNA in tumors with high CD3 presence (New Figure 4b).

These new data are now in new Figure 4 instead of the previous graphs that were partially redundant with Table1, as mentioned by the reviewer, and are now as Supplementary Figure 4.

To translate findings to a clinical setting, authors should discuss how they would propose to stratify patients, and how to administer galectin antagonists locally.

We want to stress again that the core of the paper is a proof of concept: galectins can restrict the diffusion of IFN γ in the tumor microenvironment. To answer to the reviewer's request, we have nevertheless added a new paragraph in the discussion: "This study brings the new concept that tumor-secreted lectins can diminish the availability of glycosylated soluble factors. Further studies are clearly required to test the translational potential of these findings. We already observed that human tumors with high T cell infiltration contain IFN γ hijacked by galectins, but

the efficacy of systemic administration of galectin antagonists for liberating this IFN γ has yet to be proven.”

Minor suggestions:

- Fig 3d, e: also indicate absolute CXCL9 mRNA expression (not relative to IFN γ treatment), and confirm that these levels (following IFN γ treatment) are higher with galectin mutant. In addition, authors should comment why in Fig 3e, one observes no reduction in CXCL9 mRNA with a-IFN γ R1.

Tumor sensitivity to IFN γ is highly cell line-dependent. The two parental cell lines, SKBR3 and SKBR3-GFP, reacted differently to IFN γ in terms of both CXCL9 induction and HLA-II upregulation (Annex 6). This was also true for SKBR3-Gal3-GFP and SKBR3-Gal3mut-GFP, as SKBR3-Gal3mut-GFP produced lower levels of CXCL9 and HLA-II than SKBR3-Gal3-GFP. We have no explanation for these differences, since surface expression of IFN γ receptor chains are similar (data not shown). These basal differences exist independently of galectin presence.

These cells were used in Figure 3 only to prove that galectin antagonists do not boost CXCL9 when there is no functional galectin-3. If the reviewer/editor consider that these data are not strong enough, we are ready to remove them from the manuscript.

- Fig 4 is redundant (data already present in Table 1).

We agree that Figure 4 is mostly redundant with Table 1, only contributing with the specificity controls for IFN γ (anti-IFN γ R-I antibody) and galectin antagonists (sucrose). We have moved these data to the New Supplementary Figure 4.

- Fig 5b: to better understand the contribution of lattices covering tumor cells versus those that cover collagens, it would be helpful to add the fraction of tumor content (based on HE or a tumor marker) along the tumor sections. It would also help to put figs 5 and 6 into a single figure.

We have now measured the fraction covered by tumor islets versus collagen in 26 tumor sections stained with hematoxylin-eosin. The average shows that 15 ± 4 % of the tumor is covered by collagen. This result has been added to the legend of New Supplementary Figure 8.

We would prefer not to mix figures 5 and 6 for clarity, but are ready to do it if the editor also requests it.

- Fig 7b: please represent data for tumor only (not vs spleen); and Fig 7d: please represent % and MFI separately.

Data of figures 7b and 7d have been reanalyzed and represented as proposed by the reviewer (New Figure 7b, New Supplementary Figure 9a and Annex 7). Regarding the T cell activation phenotype, we thought useful to have one number representing the global phenotype change, by multiplying the percentage of cells positive for a marker by the median fluorescence intensity of these positive cells. We thus consider the original graph more meaningful and propose to keep it in the main figure. We have no problem in showing the data like asked by the reviewer (Annex 7) in a new supplementary figure if requested.

- Fig 8: NaCLac and a-Gal3 treatments are combined, whereas these treatments are given separately in Fig 7. Please also add separate treatments in Fig 8. In addition, authors should not refer to these data as tumor regression, yet speak of delay in tumor outgrowth.

As requested, we have performed a new *in vivo* experiment analyzing tumor growth after treatment with anti-galectin-3 alone. We observed a similar delay in tumor growth using only this specific antibody (New Figure 8c). We thank the reviewer for suggesting this new control that strengthen the manuscript.

Reviewer 3 is perfectly right. We barely see any complete regression in our mouse model. We have always intended to use the term delayed tumor growth. We have checked again the manuscript to avoid the use of tumor regression.

- Discussion: please add what fraction of IFN γ normally is N-glycosylated. Also, comment on the lack of effect of galectin-3 antagonists towards chemoattractants (Fig 1d, no effect of lactose; and Fig S1); possibly unexpected since chemoattractants are glycosylated.

The IFN γ produced by CHO cells is entirely glycosylated (Annex 1). Glycosylated IFN γ shows two bands at 25 and 21 kD whereas unglycosylated IFN γ shows one band at 17 KD.

Concerning the absence of galectin-3 binding to chemokines, we were also surprised and we have no real explanation. Galectin specificity for glycans is complex, as now mentioned in the Introduction section. Thus, the glycans decorating chemokines might have low avidity moieties for galectin-3.

- Discussion: line 361, first time 'galectin-3' should be 'galectin-3 antagonists'.

We have corrected this mistake.

Annex 1. The glycosylated IFN γ samples does not contain unglycosylated forms and it is mostly captured by galectin-3 beads. a) Coomassie staining of glycosylated (produced in CHO cells) and unglycosylated IFN γ (produced in E. coli). **b)** Glycosylated IFN γ produced in CHO cells was measured by ELISA in the supernatant after incubation with galectin-3-coated beads. Each bar is the average of a duplicate.

Annex 2. CXCL9 induction in LB33-MEL incubated with IFN_γ. Fold induction CXCL9 values were calculated using HPRT-1 as reference gene and with respect to untreated condition (2^{-ΔΔCt}). Mean ± SD of triplicates.

Untreated

Glycosylated IFN γ + α Gal3 ratIgG

Unglycosylated IFN γ + ctrl ratIgG

Unglycosylated IFN γ + α Gal3 ratIgG

Days after T cell injection

Annex 3. Tumor growth in mice untreated or treated with unglycosylated or glycosylated IFN γ . n=10 for each treatment. Two-way ANOVA *** P<0.001, **** P<0.0001. Below, graphs showing the individual tumor growth for each treatment.

Annex 4. Collagenase D does not affect CXCL9 induction by IFN γ . **a)** CXCL9 induction in LB33-MEL incubated for 4 h with IFN γ (50 ng/ml), and/or collagenase D (2.5 μ g/ml). Fold induction CXCL9 values were calculated using HPRT-1 as reference gene and with respect to untreated condition ($2^{-\Delta\Delta C_t}$). Mean \pm SD of triplicates. **b)** Coomassie staining of 4 μ g/lane of collagen-I (Col-I), galectin-3 (Gal-3) and IFN γ glycosylated (produced in CHO cells) or unglycosylated (produced in *E. coli*) after 1 h incubation with 250 μ g collagenase D.

Annex 5. Ex vivo secretion of cytokines and chemokines by tumor fragments treated in vivo. Tumors were treated in vivo with IFN γ 50 ng/ml (white symbols) alone or together with 10 μ g/ml of anti-galactin-3 antibody (grey symbols). Two days later, T cells were injected through the tail vein. Four days after, tumors were extracted, cut into pieces and incubated ex vivo for another 18h. The supernatants were analyzed for several cytokines and chemokines using multiplex technology. No human IL-2, IL-4, IL-6, IL-10, IL-12, GM-CSF or CCL4 were detected. No statistically significance was observed at this time point between the two treatments.

Annex 6. Response to IFN γ differs among the different SKBR3-derived cell lines. Graphs in the left column show HLA-II upregulation 3 days after IFN γ addition (25 ng/ml) analyzed by flow cytometry. Graphs in the right column show CXCL9 mRNA induction 4 h after IFN γ addition (25 ng/ml). Each bar represents the mean \pm SD of triplicates.

Annex 7. Activation status of tumor infiltrating CD8+ T cells. Graphs show the percentage of positive cells for the different activation markers (left graph) or the median fluorescence intensity of the positive cells for these markers (right graph). Each symbol represents the mean value for each activation marker (n=6-8 mice/group). Wilcoxon matched-pairs signed rank test. P values are shown in the graphs. Representative histograms are shown in Supplementary Fig. 11.

REVIEWERS' COMMENTS:

Reviewer #1 (Remarks to the Author):

The authors have carefully considered the raised issues and have added control experiments and new data. The interpretations of the experiments have been explained in a more critical way and small text errors have been corrected. A concluding scheme has been added to illustrate the main findings. This is all well done and my recommendation is to accept the study.

Reviewer #2 (Remarks to the Author):

All concerns have been addressed.

Reviewer #3 (Remarks to the Author):

Gordon-Alonso and colleagues have addressed all concerns in a satisfactory manner; and the revised version demonstrates enhanced clarity and scientific quality.

We appreciate the kind comments made by all the reviewers (see below) and we thank them for their positive evaluation. In agreement with the reviewers, no changes or corrections have been done.

REVIEWERS' COMMENTS:

Reviewer #1 (Remarks to the Author):

The authors have carefully considered the raised issues and have added control experiments and new data. The interpretations of the experiments have been explained in a more critical way and small text errors have been corrected. A concluding scheme has been added to illustrate the main findings. This is all well done and my recommendation is to accept the study.

Reviewer #2 (Remarks to the Author):

All concerns have been addressed.

Reviewer #3 (Remarks to the Author):

Gordon-Alonso and colleagues have addressed all concerns in a satisfactory manner; and the revised version demonstrates enhanced clarity and scientific quality.